# Likelihood-based Fine-tuning of Protein Language Models for Few-shot Fitness Prediction and Design

**Alex Hawkins-Hooker**[*]
*AI Centre, University College London*

**Shikha Surana**[*]
*InstaDeep Ltd.*

**Jack Simons**[*]
*InstaDeep Ltd.*

**Jakub Kmec**
*InstaDeep Ltd.*

**Oliver Bent**
*InstaDeep Ltd.*

**Paul Duckworth**[*]
*InstaDeep Ltd.*

## Abstract

Machine learning models trained on measurements of protein functional properties are widely used to accelerate laboratory-based protein design campaigns. To maximise the signal that can be extracted from limited experimental data, sequence embeddings produced by protein language models (PLMs) are often used as the basis of supervised fitness predictors. However, embedding-based predictors do not directly exploit the distributional information encoded in PLM likelihoods after self-supervised or generative pretraining on natural protein sequences. In contrast, likelihood-based fine-tuning approaches exploit this prior knowledge by directly updating pretrained PLM likelihoods to reflect observed fitness differences between sequences. While likelihood-based fine-tuning methods have been proposed previously, a conclusive comparison of their performance against state-of-the-art embedding-based methods has been lacking. To address this gap, we conduct a comprehensive empirical evaluation of both fine-tuning strategies on a representative set of protein fitness datasets from the ProteinGym benchmark. To ensure our evaluation is applicable across different PLM classes, we develop a simple, unified framework for likelihood-based fine-tuning that applies to models trained with various objectives. Across model classes and fitness datasets, likelihood-based fine-tuning consistently outperforms embedding-based methods previously reported as state-of-the-art, with the largest gains in low-data settings. Finally, to highlight the practical relevance of these findings, we demonstrate that the best-performing fine-tuning strategies can substantially improve the maximal fitness of designed sequences in multi-round *in silico* optimisation campaigns.

## 1 Introduction

Natural protein sequences are the result of evolution via natural selection. Protein language models (PLMs) learn the distribution of natural sequences, thereby implicitly modelling functional and structural constraints relevant to protein fitness (Albanese et al., 2025). As a result, PLM likelihoods form effective zero-shot

---

[*]Work was initiated during Alex's internship at InstaDeep, and subsequently completed by the research team.

predictors of the fitness effects of mutations (Meier et al., 2021; Notin et al., 2022). These distribution learning capabilities are also highly relevant for protein sequence design: mutations assigned high likelihoods by the PLM can be iteratively incorporated to improve fitness (Hie et al., 2023), or entire sequences can be generatively sampled from the learned distributions (Sgarbossa et al., 2023; Madani et al., 2023).

In multi-round design scenarios, experimental techniques are often used to generate labelled datasets associating a collection of sequences with measurements of function-relevant biological properties. Although in some cases these properties are amenable to high-throughput experimental screening methodologies whose resulting datasets are typically large (Rocklin et al., 2017; Tsuboyama et al., 2023), in general, it might only be feasible to generate measurements for tens or hundreds of proteins in each round (Biswas et al., 2021). It is therefore of considerable interest to ask how to best leverage small labelled datasets together with the distributional knowledge of PLMs to improve fitness prediction and function-aware sequence design.

One popular paradigm for exploiting pretrained PLMs in this setting involves extracting sequence representations and utilising these as (possibly frozen) inputs into task-specific downstream predictive models (Alley et al., 2019; Biswas et al., 2021; Rao et al., 2019; Dallago et al., 2021; Khan et al., 2023; Notin et al., 2023b; Groth et al., 2024). However, trends in natural language processing have shown the benefits of directly adapting the distributions of models using task-specific labelled data or pairwise preferences (Ouyang et al., 2022; Rafailov et al., 2023) to fine-tune all parameters, thereby fully exploiting the distributional information contained in the original pretrained model. In the context of protein fitness prediction, similarly motivated pairwise ranking losses have shown promise in adapting the likelihoods of unconditional autoregressive PLMs (Krause et al., 2021; Lee et al., 2023). However, a direct head-to-head comparison with state-of-the-art approaches for exploiting frozen embeddings (Notin et al., 2023b; Groth et al., 2024), remains lacking. Moreover, the question of whether similar ranking-based strategies can effectively be applied across other popular classes of PLMs (e.g. PLMs trained with masked language modelling) is likewise relatively under-explored.

In this work, we seek to provide a thorough head-to-head comparison between likelihood-based and embedding-based fine-tuning strategies for adapting protein language models to limited labelled protein fitness datasets. To ensure the applicability of our results across protein language model classes, we develop a unified framework for likelihood-based fine-tuning, encompassing masked protein language models and family-based autoregressive models as well as simple autoregressive models through appropriate choices of the scoring function used to parameterise a probabilistic model of the relative fitnesses of pairs of sequences. We apply the resulting fine-tuning schemes to fine-tune leading masked and autoregressive family-based PLMs as supervised fitness predictors, focussing in particular on the low-data setting across a set of representative fitness datasets from ProteinGym Notin et al. (2023a). Via head-to-head comparisons with SOTA embedding-based approaches, we provide compelling empirical evidence that ranking-based fine-tuning schemes outperform embedding-based predictive strategies across various classes of PLM, especially in low-data regimes. To enhance the practical relevance of this finding, we additionally develop ensembling strategies compatible with these fine-tuning schemes, showing that they lead to further performance gains. Finally, we illustrate the effectiveness of the best-performing fine-tuning strategies across multi-round *in silico* protein optimisation campaigns.

## 2   Related work

**Zero-shot protein fitness prediction:** The most successful models for zero-shot prediction of mutation effects on protein fitness attempt to predict the likelihood of particular sets of mutations occurring within a natural protein given its evolutionary context. Traditional methods within this category involve statistical models trained directly on multiple sequence alignments for each protein of interest, such as profile models (Hopf et al., 2017), Potts models (Figliuzzi et al., 2016; Hopf et al., 2017) and VAEs (Frazer et al., 2021; Riesselman et al., 2018; Hawkins-Hooker et al., 2021a). More recent generalisations of such methods involve pretraining large PLMs across all natural proteins, adopting the paradigms of masked language modelling (Rives et al., 2021; Meier et al., 2021; Lin et al., 2022) or autoregressive modelling (Notin et al., 2022; Nijkamp et al., 2023; Madani et al., 2023; Bhatnagar et al., 2025) and evaluating the correlation of language model likelihoods with fitness effects. However, unconditional PLMs that model the distributions of individual sequences suffer from a lack of context, often requiring fine-tuning to adapt their distributions towards a particular protein family of interest Madani et al. (2023). The leading evolution-based autoregressive models

exploit additional information to address this limitation, achieving improved predictions either by biasing likelihoods with statistics from multiple sequence alignments (MSAs) (Notin et al., 2022), or by explicitly conditioning on MSAs (Hawkins-Hooker et al., 2021b; Ram & Bepler, 2022; Truong Jr & Bepler, 2023).

**Supervised protein fitness prediction:** Fitness prediction has been studied as a supervised learning task in many prior works. Several works have sought to exploit the pretrained representations of PLMs to improve performance, using either fine-tuned (Rao et al., 2019) or frozen embeddings (Dallago et al., 2021; Notin et al., 2023b; Groth et al., 2024). Nonetheless, approaches based on embeddings risk discarding useful distributional information captured in the models' output layers (Krause et al., 2021). The importance of fully leveraging distribution information for fitness prediction is highlighted by the success of 'augmented density' predictors (Hsu et al., 2022a), which combine zero-shot fitness predictions based on pretrained likelihoods with either one-hot encoded (Hsu et al., 2022a), or embedded (Notin et al., 2023b) representations of input sequences. ProteinNPT (Notin et al., 2023b) combines these strategies, training a bespoke non-parametric Transformer (Kossen et al., 2021) to reason over both zero-shot predictions and associated sequence embeddings to produce fitness predictions. Similarly, Kermut (Groth et al., 2024) introduces a composite Gaussian process kernel informed by frozen sequence embeddings and structure predictions, and models zero-shot predictions as GP prior mean function. Analogously, Metalic (Beck et al., 2025) meta-learns to condition on both unlabelled and labelled sets of sequences to improve zero or few-shot predictions for new proteins[1].

Methods seeking to adapt the distributions learned by PLMs directly have been less well studied. Rives et al. (2021) propose to use the log-likelihood ratio between mutant and wild-type amino acids as a regression function, fine-tuning the full model. Krause et al. (2021) suggest using a ranking-based loss function to fine-tune autoregressive PLMs, showing improvements over augmented density baselines on a small set of fitness datasets. A similar ranking-based loss function was proposed for training non-pretrained CNN-based architectures on fitness datasets in Brookes et al. (2023). Recently, Lee et al. (2023) apply ranking-based loss functions derived from the literature on large language model alignment (Rafailov et al., 2023) to fine-tune unconditional autoregressive PLMs. The application of ranking-based loss functions to masked PLMs is also considered in concurrent work (Zhao et al., 2024).

**Model-guided optimisation:** Another related line of work seeks to harness protein language models to improve protein design, often incorporating supervised prediction together with algorithmic strategies for selection of promising candidate sequences (e.g. Bayesian Optimisation). A brief summary of relevant work in this direction is provided in Section B.1.

## 3 Background

**Ranking-based loss functions:** We extend two recent works proposing to train protein fitness prediction models via a ranking loss based on the Bradley-Terry model (Krause et al., 2021; Brookes et al., 2023; Bradley & Terry, 1952). The Bradley-Terry model represents the probability that a given sequence $x_i$ has higher fitness $y(x_i)$ than another sequence $x_j$ by parameterising a binary classifier via the difference in scores of each sequence under a learned scoring function $s_\theta(x)$:

$$p(y(x_i) > y(x_j)) = \sigma(s_\theta(x_i) - s_\theta(x_j)), \tag{1}$$

where $\sigma$ is the logistic sigmoid function. The model is trained by maximising the likelihood of the complete set of pairwise comparisons between the fitness values of sequences with respect to the parameters $\theta$ of the scoring function. Concretely, given a batch of $B$ sequences $x_1, ..., x_B$, the loss is given by:

$$\mathcal{L} = \sum_{i=1}^{B} \sum_{j=1}^{B} -\mathbb{I}(y(x_i) > y(x_j)) \log \sigma(s_\theta(x_i) - s_\theta(x_j)), \tag{2}$$

where $\mathbb{I}$ is an indicator function. Importantly, despite this formulation's reliance on pairwise comparisons, the computational cost associated with computing the loss on a batch of $B$ inputs remains linear in $B$. This is

---

[1]An earlier version of this work was cited by Beck et al. (2025), and they make use of the part of our work dedicated to ranking-based losses for masked language models.

because only $B$ independent evaluations of the scoring function $s_\theta(x)$ are required to compute the $B \times B$ terms in the loss function in Equation 2. A reference implementation of the loss function illustrating this point is provided in Appendix A.6.

**Fine-tuning autoregressive PLMs:** To use the ranking-based loss functions to fine-tune an autoregressive protein language model, Krause et al. (2021) propose an unconditional sequence log-likelihood score function:

$$s_\theta(x) = \sum_{i=1}^{L} \log \, p(x^i | x^{<i}) \,, \tag{3}$$

where $x^i$ denotes the $i^{\text{th}}$ amino acid, and $x^{<i}$ represents the subsequence of amino acids preceding position $i$ in the sequence $x$, thereby exploiting the zero-shot scoring capability of the pretrained PLM to provide an effective initialisation for pairwise classification.

## 4 Likelihood-based fine-tuning of masked and family-based PLMs

We describe below how pairwise ranking losses can be extended to fine-tune the likelihoods of two other widely used classes of PLMs: masked language models and family-based autoregressive models. These extensions are derived from the principle that fine-tuning strategies should exploit as far as possible the properties of models that lead to strong zero-shot performance (Krause et al., 2021). We enforce this principle by suggesting an appropriate choice of the scoring function $s_\theta$ used to parameterise the Bradley-Terry model in each case.

### 4.1 Conditional scoring functions

Previous applications of ranking-based losses to fine-tune PLMs for fitness prediction have focussed on unconditional autoregressive models. However, these models often underperform when compared to other classes of models, such as conditional autoregressive models and masked language models, in fitness prediction settings Notin et al. (2023a). We therefore generalise the scoring functions introduced in the previous section to accommodate the additional conditioning information $c$ exploited by these models, resulting in conditional scoring function $s_\theta(x, c)$:

$$p\big((y(x_i) > y(x_j))|c\big) = \sigma(s_\theta(x_i, c) - s_\theta(x_j, c)). \tag{4}$$

Below, we will consider cases where $c$ represents either a wild-type sequence or a multiple sequence alignment (MSA), since conditioning on evolutionary context is especially effective in fitness prediction (Truong Jr & Bepler, 2023), however we note that the same approach could be applied to models which condition on protein structure (Hsu et al., 2022b; Widatalla et al., 2024).

### 4.2 Scoring functions for masked PLMs

Masked language models do not define a sequence-level likelihood, meaning that it is not immediately obvious how to define a scoring function for the Bradley-Terry model. Meier et al. (2021) and Johnson et al. (2024) have proposed a variety of strategies for zero-shot scoring of mutants using the likelihoods assigned to sets of mutations by masked language models. We propose to use these zero-shot scoring-functions to parameterise the Bradley-Terry model in Equation 4, allowing the models to be fine-tuned with ranking-based losses, similar to concurrent work (Zhao et al., 2024). In Section C.4 Table 7 we empirically compare a range of masked PLM scoring strategies. From this study, we choose the wild-type marginal strategy for its combination of computational efficiency and strong performance as a zero-shot scoring function.

Concretely, the 'wild-type marginals' (wt-marginals) scoring function (Meier et al., 2021) is given by the summation of the log-likelihood ratios between mutated and wild-type amino acids across mutated positions, given the unmasked wild-type sequence as input:

$$s_\theta(x, x_{\text{wt}}) = \sum_{\{i:x_{\text{wt}}^i \neq x^i\}} \log p(x^i | x_{\text{wt}}) - \log p(x_{\text{wt}}^i | x_{\text{wt}}) \,. \tag{5}$$

The wt-marginal strategy assumes an additive likelihood function over mutations, and thus may not be appropriate for multi-mutant datasets where epistasis can occur, i.e. where the combined effect of mutations at different positions is not simply the additive result of their individual effects.

## 4.3   Scoring functions for family-based PLMs

Family-based PLMs represent the conditional distribution over family members given a subset of other family members (Rao et al., 2021; Hawkins-Hooker et al., 2021b; Ram & Bepler, 2022; Truong Jr & Bepler, 2023). These models have proved effective as zero-shot fitness predictors, due to their ability to explicitly condition on evolutionary context to predict the effects of mutations.

In this paper we work with PoET (Truong Jr & Bepler, 2023), which models entire protein families autoregressively. To produce predictions given a mutant sequence $x$ and an MSA $M = \{m_1, ..., m_N\}$ of homologues of a wild-type sequence $x_{\text{wt}}$, PoET computes the likelihood of the mutant $x$ conditional on the MSA $M$. To exploit this capacity to condition on family members during fine-tuning, we condition the autoregressive scoring function in Equation 3 on the MSA sequences: $s_\theta(x, M) = \sum_{i=1}^{L} \log p(x^i | x^{<i}, M)$. Since PoET operates natively on unaligned sequences and is sensitive to alignment depth, we subsample a small set of sequences from the MSA and discard gaps before feeding them into the model, following (Truong Jr & Bepler, 2023) (details in Appendix A.1).

To increase the efficiency of fine-tuning PoET, in practice we cache a single set of hidden layer representations obtained by passing the subsampled MSA $M$ through the model, and fine-tune only the mapping between these frozen representations and the sequence likelihoods (Appendix A.2). This effectively decouples the encoding of prior context from the decoding of future amino acids.

## 4.4   Evolutionary context ensembles

Both masked and family-based autoregressive PLMs define distributions over sequence space conditioned on evolutionary context, such as a wild-type sequence or multiple sequence alignment. In the preceding sections, we effectively fixed the evolutionary context, by sampling a single input MSA with which to fine-tune family based models, and by fine-tuning masked language models using the full wild-type sequence as input. However, these models' ability to condition on evolutionary context naturally leads to the question of whether aggregating predictions across multiple related evolutionary contexts can yield improved predictions. We therefore extend our fine-tuning schemes to fine-tune ensembles of models from a single set of pretrained weights, but *conditioning on distinct evolutionary contexts*.

Concretely, we construct ensembles by repeatedly fine-tuning a single pretrained model using a different fixed perturbation of the evolutionary context (the wild-type sequence or MSA) for each fine-tuning run. For *family-based* models, we sub-sample a set of $K$ input MSAs $M_{1:K}$ from the full MSA associated with the wild-type sequence. We then fine-tune a separate set of parameters $\theta_k$ to minimise the loss conditioned on each MSA, producing $K$ sets of parameters, each specialised to a single input MSA (further details are provided in Appendix A.2). To score sequences, we define the ensemble scoring function as: $s_{\theta_{1:K}}(x, \{M_{1:K}\}) = \frac{1}{K} \sum_{k=1}^{K} s_{\theta_k}(x, M_k)$. This procedure extends the practice of MSA ensembling used to improve the zero-shot predictions of MSA-based PLMs (Truong Jr & Bepler, 2023), to the supervised learning and sequence design setting.

To apply a similar evolutionary context-based ensembling strategy to *masked* models, we sample a set of $K$ input masks, and fine-tune a separate set of parameters for each input mask, exploiting the intuition that differently masked sequences remain functionally equivalent, but may nonetheless produce different outputs when passed through the model. Further details are provided in Appendix A.3.

# 5   Low-n fitness prediction

## 5.1   Experiment Details

**Protein fitness datasets**   We study the performance of fitness prediction strategies on mutational datasets ('fitness landscapes') from ProteinGym (Notin et al., 2023a). Each dataset contains a set of protein sequences together with experimentally determined fitness values. The protein sequences within a dataset typically contain a small number of mutations relative to a natural wild-type protein, and their corresponding fitness values are quantitative measurements of a functional property associated with the wild-type. We utilise two subsets of ProteinGym: the first is the validation set of 8 representative single-mutant datasets selected by Notin et al. (2023b). The second is a set of 5 multi-mutant datasets that constitutes a non-redundant set of the most diverse landscapes available in ProteinGym (more details in Section A).

In contrast to the experiments considered in ProteinNPT and Kermut (Notin et al., 2023b; Groth et al., 2024), we focus on fine-tuning across a broad range of low-data settings. For each dataset, we train all methods on $n = 24, 32, 48, 128$ or $512$ sequences randomly sampled from the dataset and evaluate on either 2000 (single-mutants) or 5000 (multi-mutants) randomly sampled held-out sequences. All methods use an additional set of 128 randomly sampled sequences as a validation set to select the best checkpoint (i.e. for early stopping). For each dataset and each value of $n$, we generate three random train/validation/test splits and report the average test-set Spearman correlation. For models trained with ranking losses, the Spearman correlation is computed between the scoring function $s_\theta(x, c)$ and the ground truth fitness values.

Prior work has also considered non-random splits (Dallago et al., 2021; Notin et al., 2023b; Groth et al., 2024), which test the generalisation of model predictions to sequences with mutations at positions unseen in the training dataset. We note that such generalisation is already implicitly required in the low-data setting, as not all mutation positions will be present in the small training sets. That being said, in Section 5.2 we more directly assess generalisation by computing metrics on subsets of the test sets containing mutations at positions for which no mutations were present in the training set.

**Fitness prediction strategies**   We evaluate the performance of the fine-tuning strategies introduced in Section 4 on the selected ProteinGym datasets. To attain an understanding of the effectiveness of these strategies across different classes of PLM, we apply them to the masked language models ESM-1v (Meier et al., 2021) and ESM-2 (Lin et al., 2022), and the family-based autoregressive model PoET (Truong Jr & Bepler, 2023). In each case, we fine-tune all model parameters by parameterising the Bradley-Terry model of Equation 1 via the corresponding scoring functions in Section 4, and minimising the ranking loss in Equation 2.

We compare against two widely used baseline approaches: (i) fine-tuning PLMs with an added regression head (Rao et al., 2019), and (ii) training downstream predictors on frozen language model embeddings. In the first case, we add a linear regression head to pooled embeddings extracted from the models, and fine-tune with an MSE loss (further details are provided in Appendices A.1.1 and A.2). As leading examples of the second class of approaches, we compare to ProteinNPT (Notin et al., 2023b) and Kermut (Groth et al., 2024), that use frozen language model embeddings as inputs to downstream (nonparametric) regressors. Kermut results are obtained without the usage of a structure model (referred to as Kermut-sequence in (Groth et al., 2024)) to allow for fair comparisons between models.

As additional baselines, we include the 'augmented density' strategies used as baselines in Notin et al. (2023b). These are regression models which take as input the zero-shot predictions of a PLM as well as either a one-hot representation of the mutated sequence (Hsu et al., 2022a), or an embedding extracted from a PLM (further details in Appendix A.4). We refer to these distinct choices of augmented density representation as 'OHE augmented' (OHE aug.) and 'Embedding augmented' (Emb. aug.) respectively, following Notin et al. (2023b).

For all methods, we select model hyperparameters based on the supervised fine-tuning performance aggregated over the single mutant datasets for $n = 128$ data regime, in line with ProteinNPT (see Appendix A.1 for additional details).

## 5.2 Results

A full table of fitness prediction results is provided in Section C.1 Table 4 and Section C.2 Table 5. Each model is fine-tuned on the low-$n$ dataset and evaluated via Spearman correlation on the held out test set.

**Result 1: Ranking-based fine-tuning outperforms regression fine-tuning**  We first compare ranking-based and regression-based fine-tuning using identical base models across a range of low-$n$ data settings. As shown in Figure 1, we observe a consistent trend across all models: directly fine-tuning PLMs using a likelihood-based ranking loss yields higher predictive performance than regression-based fine-tuning of pooled embeddings across all tested values of $n$.

Varying the number of labelled training sequences $n$ reveals some additional patterns. At the lowest values of $n$ ranking-based fine-tuning clearly dominates. For example, ranking-based fine-tuning of ESM-1v with just 24 sequences outperforms regression fine-tuning with five times more data ($n = 128$). Similarly, for PoET, ranking-based fine-tuning with $n = 24$ outperforms the standard regression method using twice as many training sequences ($n = 48$). The performance gap narrows for all models as $n$ increases, suggesting that larger datasets partially compensate for the suboptimal initialisation introduced by adding an untrained linear prediction head to pooled embeddings.

One exception to the general trend occurs for ESM-1v in the $n = 512$ multi-mutant setting, where regression-based fine-tuning slightly outperforms ranking-based fine-tuning. We attribute this result to the additive nature of the wild-type marginal scoring rule used for masked language models, which may be insufficient to capture epistatic effects in multi-mutant settings. More expressive but computationally intensive scoring strategies may mitigate this limitation (see Section C.4 and Table 7).

To further understand the improvements introduced by ranking-based fine-tuning, in Section C.1 we attempt to disentangle the effects of the loss function from the use of the model's likelihood to initialise a scoring function. Notably, models fine-tuned with the same ranking loss but using a newly initialised linear projection applied to pooled embeddings to parametrise the scoring function often outperform standard regression models, but consistently underperform likelihood-based ranking models.

Finally, in Appendix C.7 Table 10, we provide fitness prediction results comparing the PLM ensembling strategies introduced in Section 4.4 for dataset sizes $n = 128, 512$. These results demonstrate that the proposed ensembling strategy provides a practical way to trade additional computation for improved fine-tuning performance when only a single set of base model parameters is available.

**Result 2: Ranking-based fine-tuning outperforms SOTA models trained on frozen embeddings**
We next focus on the comparison between the best-performing ranking-based fine-tuning schemes and current SOTA baselines relying on frozen embeddings from a base model: ProteinNPT, Kermut-sequence, and embedding-based 'augmented density' baselines. In Figure 2, we show that directly adapting PoET likelihoods via ranking-based fine-tuning outperforms all baseline approaches across all data settings, with the gap again largest in the lowest data regimes. Notably this performance is not simply by virtue of PoET producing better zero-shot predictions. In Section C.6, we report the zero-shot Spearman correlations from a set of relevant models, showing that the 5-model ESM-1v ensemble used by ProteinNPT (ESM-1v) outperforms PoET, the single ESM-1v model used in our fine-tuning experiments, and ESM-2. Nevertheless, even when explicitly incorporating these zero-shot predictions, ProteinNPT (ESM-1v) achieves substantially lower supervised performance compared to both PoET and ESM-2 fine-tuned via likelihood-based ranking (C.2).

**Result 3: Ranking-based fine-tuning generalises better to unseen positions**  The random splits considered in the previous sections provide an estimate of performance on held-out data. However, similar mutations can occur in both train and test sets (e.g. related amino acid substitutions at the same position), meaning that measuring performance on these test sets does not necessarily directly test a model's capacity for generalisation (Notin et al., 2023b). To address this limitation, in Table 1 we assess the capability of the fine-tuning methods to generalise to mutations at unseen positions in the test set sequences, by restricting the single mutant dataset test sets to sequences containing mutations only at positions with no mutation in the training set. Results for multi-mutant datasets are provided in Section C.3 Table 6. While there is a clear and expected drop in performance at these unseen positions, ranking-based fine-tuning directly on the

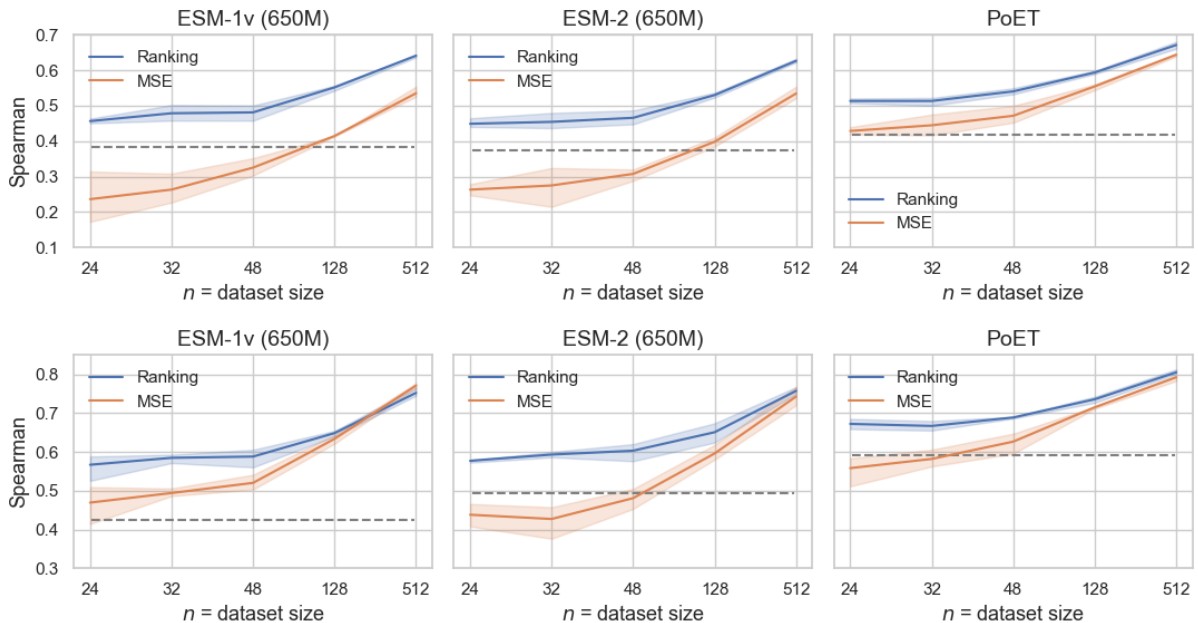

Figure 1: Supervised fitness prediction for a range of fine-tuning dataset sizes $n$ ($x$-axis) for ESM-1v (left), ESM-2 (center) and PoET (right). We directly compare Spearman correlation (higher is better) on the test set after fine-tuning using our likelihood ranking-based strategy (blue) vs the regression-based MSE strategy (orange). Zero-shot performance is represented as a dashed, gray line. In both cases all PLM parameters are fine-tuned. Average over 8 single mutant ProteinGym datasets (top), and 5 multi-mutant datasets (bottom). Error bars are $\pm\sigma$.

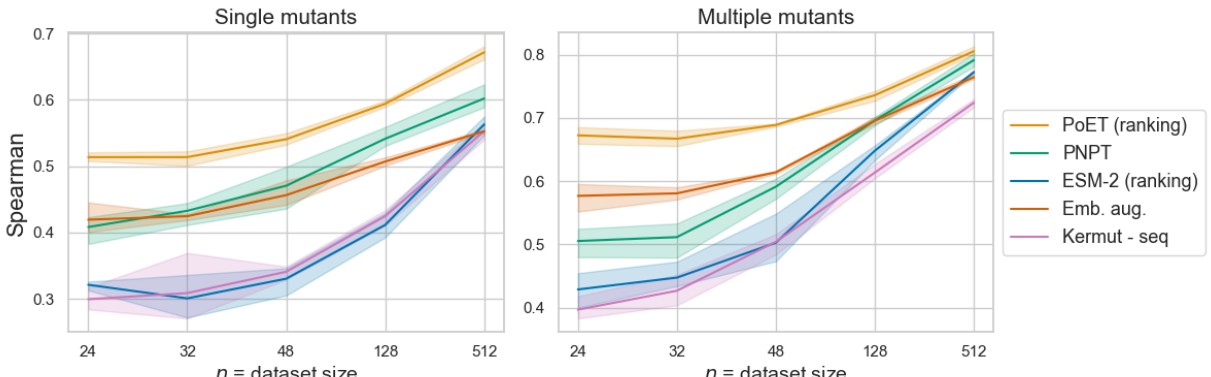

Figure 2: Supervised fitness prediction for best performing PoET and ESM-2 fine-tuning strategies over different dataset sizes $n$ ($x-$axis), against SOTA baselines ProteinNPT (MSAT), Kermut$-$seq and augmented density 'Emb aug (MSAT)'. Average Spearman correlation (higher is better) over 8 single mutant datasets (left) and 5 multi-mutant datasets (right). Error bars plotted are $\pm\sigma$.

likelihoods demonstrates the best out-of-distribution behaviour at unseen positions for all models, suggesting that the superior performance of ranking-based fine-tuning is maintained across different types of split.

**Table 1:** Spearman correlations for test set mutations at seen vs. unseen residue positions ($n$=128). Test set sequences are assigned to the unseen set if they contain mutations at residue positions at which none of the training set sequences have mutations. ProteinNPT (MSAT) (referred to as PNPT).

| Model | Scoring fn. | Loss | Seen | Unseen |
|---|---|---|---|---|
| ESM-1v | linear head | MSE | 0.460 | 0.331 |
| | wt-marginals | rank | **0.592** | **0.509** |
| ESM-2 | linear head | MSE | 0.453 | 0.297 |
| | wt-marginals | rank | **0.568** | **0.455** |
| PoET | linear head | MSE | 0.571 | 0.517 |
| | likelihood | rank | **0.612** | **0.549** |
| PNPT | - | MSE | 0.563 | 0.462 |
| Kermut-seq | GP head | NLL | 0.464 | 0.363 |

## 6 Multi-round optimisation

### 6.1 Experiment detailss

We next ask whether the improvements in fitness prediction translate to benefits in multi-round sequence design experiments. We follow a similar multi-round sequence design setting to that introduced by Notin et al. (2023b): sequence design is formulated as a pool-based optimisation task over the sequences in an experimental fitness dataset. For a given dataset, the goal is to retrieve as many high-scoring sequences as possible in 10 optimisation rounds. In each round, the model is fine-tuned using labelled sequences collected across previous rounds. The model's predictions are then used to guide the acquisition of a batch of $N$ sequences from a pool of candidate sequences drawn from the ground truth dataset (for multi-mutant datasets, we limit the pool size to 5000 sequences). More precisely, in each round we rank all remaining sequences in the pool by the values of an *acquisition function* derived from the model's predictions, and select the top $N$ to add to the training set. For single models, we use a greedy acquisition strategy, where the acquisition function corresponds directly to the predicted fitness (for regression-based models) or the learned scoring function (for ranking-based models). When an ensemble of fitness prediction models is available, we instead use an uncertainty-aware acquisition function. For consistency with (Notin et al., 2023b), we use the upper confidence bound (UCB) acquisition function with $\lambda = 0.1$ in all cases (Section A.5), and defer a more detailed comparison of acquisition functions suitable for use with ensembles of fine-tuned PLMs to future work. For ProteinNPT baselines, we use the same acquisition function, and use Monte Carlo dropout (Gal & Ghahramani, 2016) to produce uncertainty estimates. We initialise all models with 100 sequences randomly sampled from the dataset. All experiments are run on three random seeds. Following (Notin et al., 2023b), we evaluate performance using recall of the top 100 ground-truth sequences among all sequences retrieved during the design campaign.

### 6.2 Results

**Result 4: Ranking-based fine-tuning improves multi-round sequence design** Figure 3 compares the proportion of high fitness sequences retrieved by iteratively fine-tuned PLMs across a range of acquisition batch sizes ($N$) on the single mutant datasets. Using a ranking loss consistently improves the fitness of retrieved sequences compared to fine-tuning the same base model with a regression loss, ultimately leading to the acquisition of more high value candidates earlier in the design campaign. Notably, PoET fine-tuned with a ranking-based loss can achieve comparable performance to a regression method with significantly fewer sequences. For example, PoET achieves over 40% recall of the top 100 sequences after 10 rounds with $N = 24$, whereas a regression-based approach requires 10 rounds with $N = 64$ to achieve similar performance. For each model, ensembling multiple predictors leads to modest further improvements (Appendix C.8 and Table 11).

For direct numerical comparison with ProteinNPT, we report the area under top 100 recall curve (AUC) for the best performing fine-tuning method associated with each model in Table 2. AUCs for all methods

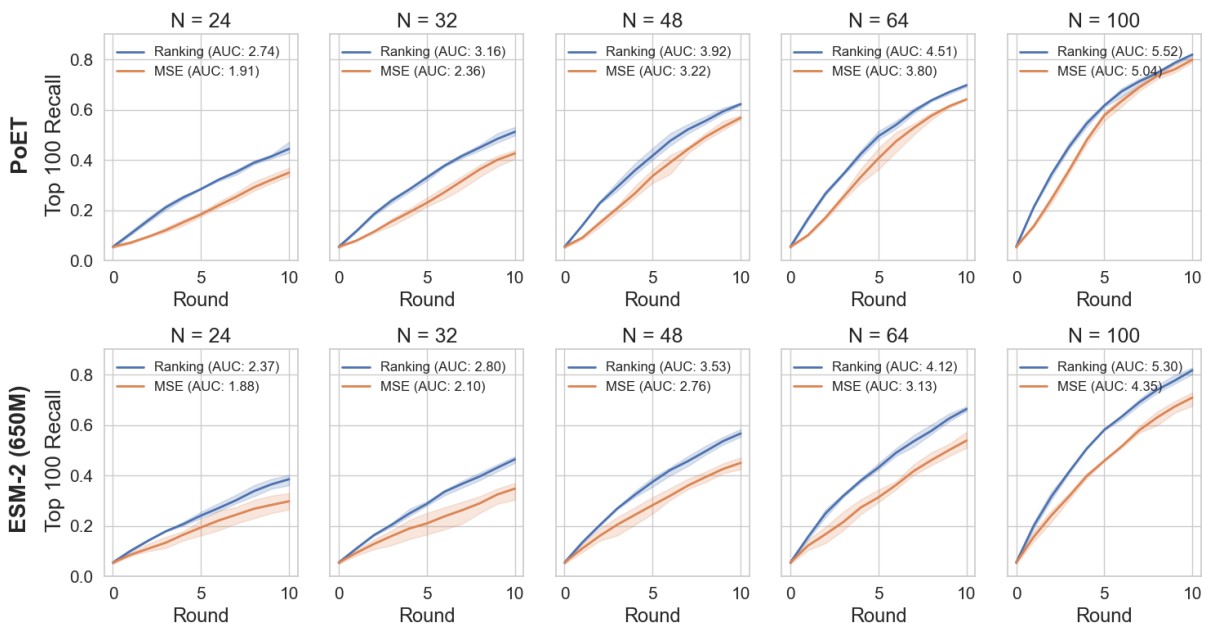

Figure 3: Multi-round sequence design for a range of acquisition batch sizes $N$ over 10 rounds ($x-$axis) for PoET (top) and ESM-2 (bottom). We compare recall of the top 100 sequences from the ground truth ProteinGym datasets (higher is better) after fine-tuning using our likelihood ranking-based strategy (blue) or the regression- based MSE strategy (orange). In both cases all PLM parameters are fine-tuned. Lines show average over 8 single mutant datasets. Error bars are $\pm\sigma$ (across 3 random seeds).

Table 2: Multi-round sequence design recall AUC metric (higher is better) for acquiring the top 100 sequences in the candidate pool. Averaged over 8 single mutant datasets and 5 multi-mutant datasets. Full recall curves are provided in Appendix C.8 Figure 4, 5, and 6.

| Model Name | Scoring Fn. | Loss | Singles | Multiples |
|---|---|---|---|---|
| ESM-2 Ensemble | linear head | mse | 4.57 (0.11) | 3.34 (0.23) |
| | wt-marginals | **ranking** | **5.30** (0.06) | **3.91** (0.14) |
| PoET MSA Ensemble | linear head | mse | 5.11 (0.02) | 4.29 (0.20) |
| | likelihood | **ranking** | **5.59** (0.05) | **4.45** (0.28) |
| ProteinNPT + MC dropout | - | mse | 5.57 (0.02) | 4.12 (0.13) |

are provided in Appendix Table 11, together with corresponding averaged and per-dataset design curves in Appendices C.8, Section C.8.4 (singles datasets) and Section C.8.5 (multiples datasets). The relative ordering of methods is largely stable across individual datasets, although in some cases non-PLM baselines perform comparably to the best-performing methods, suggesting that these datasets may contain noisy or otherwise difficult-to-predict fitness labels (Notin et al., 2023b). Interestingly, these curves and their AUC summaries indicate that greedy batch selection with individual fine-tuned models performs almost as well as ensemble-based acquisition strategies. We hypothesise that this is due either to poorly calibrated uncertainty estimates or to properties of the tasks themselves that favour a greedy approach; we leave further investigation of this phenomenon to future work.

## 7   Conclusion

In this paper, we study likelihood-based fine-tuning strategies for protein language models and show how they can be applied effectively across both masked language models and family-based autoregressive architectures

through appropriate choices of scoring functions to parameterise a ranking model. We demonstrate that these likelihood scoring functions enable effective ranking-based fine-tuning, leading to significant improvements in fitness prediction over zero-shot methods, from as few as 24 experimental measurements, comparable to typical batch sizes in biological experiments. Across both supervised prediction and multi-round design tasks, likelihood-based finetuning consistently outperforms leading baselines, including ProteinNPT and Kermut. Together, these results provide convincing empirical evidence that ranking-based fine-tuning should be preferred over regression-based alternatives, regardless of PLM architecture, particularly in low-data regimes. Finally, we introduce ensembling strategies that are compatible with likelihood-based fine-tuning and show that they further improve performance in both single and multi-round settings. Overall, our results provide a solid empirical foundation for future work on protein language model-assisted protein design with laboratory feedback. Natural extensions include studying the generative capabilities of PLMs fine-tuned with likelihood-based losses, as well as further investigations into methods for uncertainty quantification building on state-of-the-art fine-tuning strategies.

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

# A   Additional experimental details

We use the set of 8 single-mutant datasets selected for ablations and hyperparameter selection by (Notin et al., 2023b). The names of these datasets in ProteinGym are:

- BLAT_ECOLX_Jacquier_2013
- CALM1_HUMAN_Weile_2017
- DYR_ECOLI_Thompson_2019
- DLG4_RAT_McLaughlin_2012
- REV_HV1H2_Fernandes_2016
- TAT_HV1BR_Fernandes_2016
- RL40A_YEAST_Roscoe_2013
- P53_HUMAN_Giacomelli_WT_Nutlin

We additionally select a set of 5 of the most diverse multi-mutant datasets in ProteinGym. To select these datasets, we identified the datasets with the largest number of mutations in ProteinGym, and discarded redundant datasets (for example the GFP datasets of (Gonzalez Somermeyer et al., 2022) are datasets of close homologues of the GFP protein whose dataset was reported by (Sarkisyan et al., 2016). We therefore include only the latter.

The selected multi-mutant datasets are:

- PABP_YEAST_Melamed_2013
- CAPSD_AAV2S_Sinai_2021
- GFP_AEQVI_Sarkisyan_2016
- GRB2_HUMAN_Faure_2021
- HIS7_YEAST_Pokusaeva_2019

Table 3: Additional details of the 8 single mutation, and 8 multi-mutation datasets we use from ProteinGym. wt-length is the number residues in the wild-type protein. Mutations refers to whether the dataset contains single mutant sequences, or a range of the number of mutations present in the dataset. Fitness score distribution is provided as mean and std.

| Dataset Name | wt length | Mutations | Num of seqs | Fitness Score Distr. |
|---|---|---|---|---|
| BLAT_ECOLX_Jacquier_2013 | 286 | Single | 989 | $-1.558 \pm 1.952$ |
| CALM1_HUMAN_Weile_2017 | 149 | Single | 1,813 | $0.742 \pm 0.365$ |
| DLG4_RAT_McLaughlin_2012 | 724 | Single | 1,576 | $-0.172 \pm 0.406$ |
| DYR_ECOLI_Thompson_2019 | 159 | Single | 2,363 | $-0.391 \pm 0.742$ |
| P53_HUMAN_Giacomelli_2018_WT_Nutlin | 393 | Single | 7,467 | $-0.020 \pm 1.035$ |
| REV_HV1H2_Fernandes_2016 | 116 | Single | 2,147 | $-0.121 \pm 0.218$ |
| RL40A_YEAST_Roscoe_2013 | 128 | Single | 1,195 | $-0.265 \pm 0.345$ |
| TAT_HV1BR_Fernandes_2016 | 86 | Single | 1,577 | $-0.116 \pm 0.197$ |
| CAPSD_AAV2S_Sinai_2021 | 735 | Multiple (1-28) | 42,328 | $-1.226 \pm 3.045$ |
| GFP_AEQVI_Sarkisyan_2016 | 238 | Multiple (1-15) | 51,714 | $2.658 \pm 1.059$ |
| GRB2_HUMAN_Faure_2021 | 217 | Multiple (1-2) | 63,366 | $-0.793 \pm 0.467$ |
| HIS7_YEAST_Pokusaeva_2019 | 220 | Multiple (1-28) | 496,137 | $0.619 \pm 0.449$ |
| PABP_YEAST_Melamed_2013 | 577 | Multiple (1-2) | 37,708 | $0.524 \pm 0.391$ |

## A.1   Hyperparameter details

Hyperparameters for the fine-tuning methods were selected based on performance on the single mutant set, consistent with the practice used to select hyperparameters for the baselines from ProteinNPT. We report

metrics obtained when using these hyperparameters on both single-mutant and multiple-mutant datasets for each method.

ESM-1v, ESM-2 and PoET models were fine-tuned using the Adam optimizer (Kingma & Ba, 2015) using gradient accumulation with an effective batch size of 32. Learning rates for regression-based and ranking-based fine-tuning were selected separately in each case after after a sweep over the values `1e-4,3e-5,1e-5` on the 8 single mutant datasets. For ESM models, we computed the loss by scoring all sequences using the wt-marginal strategy. In the fitness prediction experiments, the models were trained for 50 epochs. During training on each dataset the Spearman correlation, computed on a separate validation set of 128 sequences from the dataset, was used to determine the epoch whose checkpoint should be used to produce predictions on the test set.

### A.1.1 Regression heads

Linear regression heads were added to embeddings extracted from ESM-1v, ESM-2 and PoET. In the former case, we averaged embeddings across the sequence dimension before inputting them to the regression head, in the latter case we used final token embeddings.

### A.1.2 Ensembles

Contextual ensemble models of size 5 were used for both ESM-2 and PoET. During design, the ensemble members were trained for 20 epochs each round. In each round all ensemble members were reinitialised from the pretrained model and retrained on the latest training dataset.

### A.2 PoET: more details

**Decoder-only fine-tuning**   PoET parameterises a sequence of conditional distributions over the amino acids in a set of protein sequences in the same family. The model represents the joint likelihood of a set of sequences $M = \{m_1, ..., m_N\}$, via an autoregressive factorisation over sequences and over positions within each sequence:

$$p(M) = \prod_i p(m_i | m_{<i}) = \prod_{i,j} p(m_i^j | m_i^{<j}, m_{<i}) \,. \tag{6}$$

To parameterise this distribution, PoET uses a causally masked Transformer architecture, which maps from previous amino acids to logits for the current amino acid. Conceptually, this function can be decomposed into two stages: first the entire history of previous sequences $m_{<i}$ is encoded into a sequence of embeddings $H_{<i} \in \mathbb{R}^{L \times (i-1) \times D \times E}$, where $D$ is the number of layers and $E$ is the embedding dimension, via a stack of causally masked layers:

$$H_{<i} = f_\theta(m_{<i}) \,. \tag{7}$$

The current sequence $m_i$ is then decoded by a function which maps these prior sequence embeddings and previous amino acids in the current sequence to logits for each position $j$:

$$\text{logit}_{ij} = g_\theta(m_i^{<j}, H_{<i}) \,. \tag{8}$$

To fine-tune PoET from fitness data, we propose to fine-tune only the weights of the function $g$, representing the 'decoding' of the current sequence given its context. To achieve this, we first clone the PoET weights, producing a set of 'encoder' weights $\phi$ and a set of 'decoder' weights $\theta$. We use the frozen encoder weights to produce an embedding $H \in \mathbb{R}^{L_M \times D \times E}$ of the input MSA sequences: $H = f_\phi(\{m_1, ..., m_N\})$, where $L_M$ is the total length of all sequences in the input MSA. We then fine-tune the weights $\theta$ of the cloned 'decoder' to minimise the cross-entropy loss of Equation 2 on the labelled data. Concretely, the scoring function used to parameterise the Bradley-Terry model becomes:

$$s_\theta(x, M) \equiv s_\theta(x, H) = \sum_i \log p_\theta(x^i | x^{<i}, H) \tag{9}$$

To maximise computational efficiency, the MSA embeddings $H$ are pre-computed before the start of the fine-tuning process, and remain frozen throughout.

**PoET MSA subsampling**  For PoET, in both single-model and ensemble configurations, we sampled context sequences from the same filtered MSAs used to extract MSA Transformer embeddings for ProteinNPT. These MSAs are generated from the full MSAs provided with ProteinGym by running hhfilter (Steinegger et al., 2019), requiring a minimum coverage of 75% and a maximum sequence identity of 90%. Subsequently, we use weighted sampling as described in Truong Jr & Bepler (2023) to select sequences to pass as context to PoET, up to a maximum context length of 8192 tokens. The MSA is encoded using a frozen copy of the PoET model into a set of cached hidden representations, as described in Appendix A.2. When ensembling, a separate MSA is sampled for each ensemble member, and held fixed during the fine-tuning of that ensemble member.

### A.3   ESM-1v and ESM-2: more details

**ESM-2 ensembling strategy**  To fine-tune an ensemble of models given a single ESM-2 checkpoint, we randomly sampled a set of 5 masks. Within each mask, each sequence position had a 15% probability of being masked. We fine-tuned one model for each mask, by using the correspondingly masked wild-type sequence $\tilde{x}_{\text{wt}}^{(k)}$ as input to the model, instead of the unmasked wild-type sequence. The ensembled scoring function used to generate predictions is:

$$s(x, x_{\text{wt}}) = \frac{1}{K} \sum_k s(x, \tilde{x}_{\text{wt}}^{(k)}). \tag{10}$$

### A.4   Baseline models

ProteinNPT, the embeddings augmented (Emb. aug.) baselines, and the one-hot encoding augmented (OHE aug.) baselines were all run using the code released by Notin et al. (2023b). The one-hot and embedding augmented models both use the strategy from Hsu et al. (2022a), combining the zero-shot predictions from a pretrained model with sequence features in a regression framework. They differ in the way sequence features are extracted: in the former case, ridge regression is performed directly on the one-hot encoded sequences. In the latter case, PLM embeddings are used to featurise the sequences. We refer to Notin et al. (2023b) for further details.

For the fitness prediction experiments, separate ProteinNPT models were trained for 2000 and 10000 steps, and the results of the best-performing model were reported. The other baselines appeared to benefit more from longer training and were trained for 10000 steps, as in Notin et al. (2023b). For design experiments, we used the Monte Carlo dropout uncertainty quantification strategy proposed by Notin et al. (2023b) for both ProteinNPT and other baselines. Notin et al. (2023b) report best results with a 'hybrid' uncertainty quantification strategy, however this strategy is not implemented in the publicly available code.

### A.5   Upper confidence bound acquisition function

For all ensembles, we use a parameter of $\lambda = 0.1$ within the upper confidence bound acquisition function $a(x, \lambda) = \mu(x) + \lambda\sigma(x)$, where $\mu(x)$ and $\sigma(x)$ respectively are the mean and standard deviation of the predictions of the model given an input sequence $x$.

### A.6   Implementation of ranking loss

We implement a pairwise ranking loss by using binary cross-entropy to train the model on ranking tasks by comparing predicted scores for all pairs in a batch. It computes the difference in predictions for every pair, determines the ground-truth ranking based on actual values, and applies BCE loss to these pairwise

comparisons, excluding self-comparisons via a diagonal mask. An example PyTorch implementation can be found below:

```python
import torch
import torch.nn.functional as F

def full_ranking_bce(preds: torch.Tensor, targets: torch.Tensor) -> torch.Tensor:
    # Calculate pairwise differences between all predictions
    pairwise_diffs = preds[:, None] - preds[None, :]

    # Determine if each target is greater than others in a pairwise manner
    target_comparisons = targets[:, None] > targets[None, :]

    # Compute binary cross-entropy loss for pairwise comparisons
    ranking_loss = F.binary_cross_entropy_with_logits(pairwise_diffs,
        target_comparisons.float(), reduction='none')

    # Create a mask to exclude diagonal elements (self-comparisons)
    batch_size = preds.size(0)
    diag_mask = 1 - torch.eye(batch_size, device=preds.device)

    # Apply the mask and calculate the mean loss, excluding the diagonal
    masked_loss = 0.5 * ranking_loss * diag_mask
    return masked_loss.mean((-1, -2))
```

## B  Protein Design Related Work

### B.1  Model-guided protein design

Several works have proposed variants of Bayesian optimization (BO) for designing biological sequences, including proteins (Gruver et al., 2021; Jain et al., 2022; Khan et al., 2023; Stanton et al., 2022; Hie & Yang, 2022; Amin et al., 2025). It is common to evaluate BO approaches in an unconstrained setting, where sequences are proposed by an optimiser and evaluated with a black-box oracle. However, recent work suggests that designing such biological oracles is a challenging task in itself (Buttenschoen et al., 2024; Surana et al., 2024). An alternative *in silico* evaluation strategy avoids the challenge of defining a meaningful oracle function by adopting a pool-based optimisation problem formulation over experimentally determined fitness datasets (Notin et al., 2023b). Another line of work has sought to provide direct experimental validation of approaches combining uncertainty estimates with PLMs, in settings ranging from zero- (Hie et al., 2023) to few-shot design (Biswas et al., 2021), to single-round design with large training sets of sequence-fitness pairs (Li et al., 2023).

Gruver et al. (2021) study various choices of surrogate model for protein design with BO, finding CNN ensembles to be particularly robust to the kinds of distribution shift encountered during online design. More recently, Greenman et al. (2025) studied a range of uncertainty quantification strategies applied to models trained directly on sequences, and on frozen language model embeddings.

### B.2  Relationship to preference learning for LLMs

Direct preference optimisation (DPO) (Rafailov et al., 2023) is a recently proposed method for aligning large language models (LLMs) using datasets of human preference data. DPO also uses scoring functions from pretrained models to parameterise a Bradley-Terry model. Instead of parameterising a classifier directly via differences in log likelihoods, DPO uses the difference in scaled log likelihood *ratios* between the fine-tuned model and a frozen reference model. Thus the probability that a completion $x_1$ is preferred to a completion $x_2$ given an input prompt $c$ is modelled as:

$$p_\theta(x_1 \succ x_2 | c) = \sigma\left(\beta \log \frac{p_\theta(x_1|c)}{p_{\text{ref}}(x_1|c)} - \beta \log \frac{p_\theta(x_2|c)}{p_{\text{ref}}(x_2|c)}\right). \tag{11}$$

In our notation, the DPO preference model therefore amounts to a particular choice of scoring function $s_\theta(x,c) = \beta\log\frac{p(x|c)}{p_{\text{ref}}(x|c)}$. Assuming an autoregressive decomposition of $p(x|c)$, this scoring function is equivalent to the conditional autoregressive scoring function,

$$s_\theta(x,c) = \sum_{i=1}^{L} \log p(x^i|x^{<i}, c) \,, \tag{12}$$

if the reference model is constant and $\beta = 1$.

The non-constant reference model in DPO imposes a KL penalty on the deviation between the fine-tuned $p_\theta$ and the reference model, which helps prevent collapse in the fine-tuned distribution (Rafailov et al., 2023). Although some recent work has reported success in adapting DPO to the protein fitness prediction setting (Lee et al., 2023), in our own experiments we did not find this regularisation necessary to achieve good performance. We hypothesise that this is because we do not require generations from the model, unlike typical applications of DPO.

## C   Additional Results

### C.1   Fitness Prediction: Score function and Loss Ablation

A core contribution of our work is the application of ranking-based loss function to directly fine-tune the likelihoods of masked (ESM-1v and ESM-2) and conditional autoregressive (PoET) PLM models. As an additional ablation study, we apply i) the ranking loss function in Equation (2) to the setting where a linear regression head is applied to the model embeddings, and ii) an MSE loss function is applied to the likelihood scoring functions. Below is the full table of results for each of the four settings.

Whilst we notice that the ranking loss function applied to the regression setting performs quite well, the MSE loss directly applied to the likelihood scoring functions does not.

Table 4: Low-n fitness prediction comparing masked and family-based PLM scoring strategies and loss functions on the Spearman correlation (higher is better). Evaluated on 5 multi-mutant datasets from ProteinGym. (Standard deviation over three seeds in parenthesis).

| Model | Scoring Fn. | Loss | $n = 24$ | $n = 32$ | $n = 48$ | $n = 128$ | $n = 512$ |
|---|---|---|---|---|---|---|---|
| **Single Mutants** | | | | | | | |
| ESM-1v | linear head | mse | 0.236 (0.07) | 0.263 (0.04) | 0.325 (0.02) | 0.415 (0.00) | 0.535 (0.02) |
| (650M) | | ranking | 0.312 (0.01) | 0.326 (0.05) | 0.353 (0.02) | 0.437 (0.02) | 0.590 (0.01) |
| | wt-marginals | mse | 0.300 (0.01) | 0.301 (0.01) | 0.246 (0.00) | 0.282 (0.02) | 0.495 (0.01) |
| | | **ranking** | **0.457** (0.01) | **0.479** (0.02) | **0.481** (0.02) | **0.552** (0.01) | **0.641** (0.00) |
| ESM-2 | linear head | mse | 0.263 (0.02) | 0.275 (0.06) | 0.308 (0.02) | 0.398 (0.01) | 0.535 (0.02) |
| (650M) | | ranking | 0.321 (0.01) | 0.301 (0.03) | 0.330 (0.02) | 0.411 (0.02) | 0.563 (0.01) |
| | wt-marginals | mse | 0.298 (0.01) | 0.330 (0.02) | 0.258 (0.03) | 0.267 (0.03) | 0.475 (0.01) |
| | | **ranking** | **0.449** (0.01) | **0.455** (0.02) | **0.466** (0.02) | **0.530** (0.01) | **0.627** (0.01) |
| PoET | linear head | mse | 0.429 (0.01) | 0.445 (0.03) | 0.472 (0.02) | 0.555 (0.01) | 0.644 (0.00) |
| | | ranking | 0.456 (0.03) | 0.468 (0.01) | 0.502 (0.03) | 0.575 (0.00) | 0.664 (0.01) |
| | likelihood | mse | 0.416 (0.01) | 0.409 (0.01) | 0.403 (0.01) | 0.378 (0.01) | 0.230 (0.03) |
| | | **ranking** | **0.513** (0.01) | **0.514** (0.01) | **0.541** (0.01) | **0.594** (0.00) | **0.672** (0.01) |
| Kermut | GP head | NLL | 0.299 (0.02) | 0.309 (0.05) | 0.341 (0.01) | 0.425 (0.01) | 0.553 (0.01) |
| **Results with LoRA** | | | | | | | |
| ESM-2 | linear head | mse | 0.266 | 0.295 | 0.321 | 0.417 | 0.520 |
| (650M) | wt-marginals | **ranking** | **0.400** | **0.407** | **0.435** | **0.517** | **0.616** |
| **Multiple Mutants** | | | | | | | |
| ESM-1v | linear head | mse | 0.469 (0.05) | 0.494 (0.01) | 0.520 (0.02) | 0.634 (0.01) | 0.771 (0.01) |
| (650M) | | ranking | 0.438 (0.02) | 0.474 (0.02) | 0.524 (0.02) | 0.644 (0.00) | **0.777** (0.01) |
| | wt-marginals | mse | 0.459 (0.02) | 0.451 (0.01) | 0.411 (0.01) | 0.402 (0.02) | 0.544 (0.00) |
| | | **ranking** | **0.567** (0.04) | **0.585** (0.01) | **0.588** (0.03) | **0.649** (0.01) | 0.753 (0.01) |
| ESM-2 | linear head | mse | 0.438 (0.03) | 0.427 (0.04) | 0.481 (0.03) | 0.596 (0.02) | 0.743 (0.02) |
| (650M) | | ranking | 0.429 (0.03) | 0.447 (0.02) | 0.503 (0.04) | 0.648 (0.01) | **0.773** (0.00) |
| | wt-marginals | mse | 0.454 (0.02) | 0.461 (0.01) | 0.345 (0.02) | 0.407 (0.00) | 0.522 (0.02) |
| | | **ranking** | **0.577** (0.00) | **0.593** (0.01) | **0.603** (0.02) | **0.651** (0.02) | 0.758 (0.01) |
| PoET | linear head | mse | 0.558 (0.04) | 0.582 (0.03) | 0.627 (0.03) | 0.715 (0.00) | 0.793 (0.01) |
| | | ranking | 0.571 (0.00) | 0.574 (0.02) | 0.632 (0.01) | 0.723 (0.01) | 0.801 (0.00) |
| | likelihood | mse | 0.600 (0.00) | 0.601 (0.00) | 0.600 (0.01) | 0.583 (0.00) | 0.424 (0.01) |
| | | **ranking** | **0.672** (0.01) | **0.667** (0.01) | **0.689** (0.00) | **0.736** (0.01) | **0.805** (0.01) |
| Kermut | GP head | NLL | 0.397 (0.02) | 0.426 (0.02) | 0.504 (0.02) | 0.614 (0.01) | 0.724 (0.01) |

## C.2 Fitness Prediction: ProteinNPT Baseline methods

Similarly, we ablate fine-tuning the ProteinNPT model, with both ESM-1v or MSAT frozen embeddings, using the MSE loss, as proposed in (Notin et al., 2023b), and also with the ranking loss function in Equation (2). We see that the ranking loss function improves the results across all the single mutant datasets, but results are mixed for the multi-mutant datasets.

ProteinNPT baselines (Notin et al., 2023b) that utilize frozen embeddings. Spearman correlation (higher is better) on 8 single mutant datasets and 5 multi-mutant datasets from ProteinGym.

Table 5: Low-n fitness prediction for ProteinNPT baselines (Notin et al., 2023b) that utilize frozen embeddings. Spearman correlation (higher is better) on 8 single mutant datasets and 5 multi-mutant datasets from ProteinGym.

| Model Name | Frozen Emb. | $n = 24$ | $n = 32$ | $n = 48$ | $n = 128$ | $n = 512$ |
|---|---|---|---|---|---|---|
| **Single Mutants** | | | | | | |
| ProteinNPT | MSAT | 0.408 (0.02) | 0.433 (0.02) | 0.470 (0.03) | 0.541 (0.02) | 0.602 (0.02) |
| | ESM-1v | 0.376 (0.03) | 0.415 (0.02) | 0.428 (0.02) | 0.489 (0.01) | 0.60 (0.02) |
| Emb. aug. | MSAT | 0.419 (0.02) | 0.424 (0.01) | 0.456 (0.02) | 0.507 (0.01) | 0.553 (0.0) |
| | ESM-1v | 0.446 (0.01) | 0.451 (0.01) | 0.477 (0.01) | 0.505 (0.0) | 0.550 (0.0) |
| OHE aug. | MSAT | 0.418 (0.01) | 0.429 (0.01) | 0.441 (0.0) | 0.467 (0.01) | 0.496 (0.0) |
| | ESM1v | 0.457 (0.01) | 0.466 (0.01) | 0.478 (0.0) | 0.502 (0.01) | 0.526 (0.0) |
| OHE | - | 0.116 (0.01) | 0.144 (0.03) | 0.179 (0.01) | 0.314 (0.01) | 0.488 (0.0) |
| Kermut-seq | ESM-2 | 0.299 (0.02) | 0.309 (0.05) | 0.341 (0.01) | 0.425 (0.01) | 0.553 (0.01) |
| **Multiple Mutants** | | | | | | |
| ProteinNPT | MSAT | 0.505 (0.02) | 0.511 (0.03) | 0.591 (0.02) | 0.696 (0.01) | 0.791 (0.01) |
| ProteinNPT | ESM-1v | 0.425 (0.07) | 0.438 (0.05) | 0.501 (0.03) | 0.640 (0.01) | 0.769 (0.0) |
| Emb. aug. | MSAT | 0.577 (0.02) | 0.581 (0.01) | 0.614 (0.0) | 0.696 (0.01) | 0.764 (0.0) |
| Emb. aug. | ESM1v | 0.455 (0.02) | 0.440 (0.01) | 0.505 (0.02) | 0.624 (0.01) | 0.702 (0.01) |
| OHE aug. | MSAT | 0.607 (0.01) | 0.616 (0.01) | 0.622 (0.01) | 0.684 (0.01) | 0.763 (0.01) |
| OHE aug. | ESM1v | 0.443 (0.01) | 0.460 (0.01) | 0.475 (0.02) | 0.566 (0.02) | 0.711 (0.01) |
| OHE | | 0.255 (0.02) | 0.268 (0.02) | 0.314 (0.02) | 0.473 (0.01) | 0.664 (0.01) |
| Kermut | ESM-2 | 0.397 (0.02) | 0.426 (0.02) | 0.504 (0.02) | 0.614 (0.01) | 0.724 (0.01) |

## C.3   Additional Results: Generalisation of Seen vs Unseen residue positions

We provide Spearman correlation results for the $n = 128$ fitness prediction setting specifically looking at out-of-distribution generalisation at unseen mutated residues. That is, for single mutant datasets the number of *seen* mutant positions in the training datasets (varies per dataset): min=60, max=114, mean=82.67, and the number of *unseen* mutant positions: min=17, max=657, mean=172.46. For single-mutant datasets, we classify test set sequences as seen (mutation present in the train set) or unseen (mutation absent). For multi-mutant datasets, a test set sequence is considered seen if it has up to two mutations that occur in the train set sequences; otherwise, it is unseen. The number of seen sequences in the test set of multi-mutant datasets is on average 2,124 (equivalent to approximately 42% of sequences in the test set) and 2,875 unseen sequences. Note, the test set size varies per protein dataset, however, for multi-mutant datasets we limit it to 5,000 sequences.

Table 6: Seen vs Unseen Spearman correlation scores (higher is better) evaluated on the 8 single mutant datasets (left) and 5 multi-mutant datasets (right) for the $n = 128$ dataset setting.

| | | | single-mutants | | multi-mutants | |
| Model Name | Scoring Function | Loss Type | Seen | Unseen | Seen | Unseen |
|---|---|---|---|---|---|---|
| ESM-1v | linear head | mse | 0.460 | 0.331 | 0.646 | 0.604 |
| (650M) | | ranking | 0.492 | 0.315 | 0.651 | 0.609 |
| | wt-marginals | mse | 0.350 | 0.234 | 0.412 | 0.387 |
| | | ranking | **0.592** | **0.509** | 0.652 | 0.643 |
| ESM-2 | linear head | mse | 0.453 | 0.297 | 0.605 | 0.556 |
| (650M) | | ranking | 0.447 | 0.335 | 0.649 | **0.622** |
| | wt-marginals | mse | 0.329 | 0.192 | 0.423 | 0.385 |
| | | ranking | **0.568** | **0.455** | **0.658** | 0.620 |
| PoET | linear head | mse | 0.571 | 0.517 | 0.700 | 0.695 |
| | | ranking | 0.601 | 0.535 | 0.716 | 0.715 |
| | likelihood | mse | 0.382 | 0.366 | 0.576 | 0.613 |
| | | ranking | **0.612** | **0.549** | **0.728** | **0.741** |
| Kermut | GP head | NLL | 0.464 | 0.363 | 0.607 | 0.558 |
| ProteinNPT (MSAT) | - | mse | 0.563 | 0.462 | 0.694 | 0.670 |
| | - | ranking | 0.579 | 0.474 | 0.675 | 0.664 |
| ProteinNPT (ESM-1v) | - | mse | 0.529 | 0.420 | 0.641 | 0.601 |
| | - | ranking | 0.553 | 0.465 | 0.642 | 0.607 |

## C.4   Additional Results: More Expressive Masked Scoring Functions

In Table 7 we show the complete results for additional masked PLM scoring functions that attempt to capture the epistasis effects in the multi-mutant datasets. We provide results for ESM-1v and ESM-2 for the additional strategies applied to five multi-mutant datasets, as introduced in Meier et al. (2021) and Johnson et al. (2024).

Note, due to GPU memory requirement of the masked-modulo strategy, we reduce the hyperparameters relative to the length of the protein sequence in order to fit on an H100 GPU. For example, batch size ($B$) and $K$ for each dataset were set to GRB: ($B$=8, $K$=8), GFP: ($B$=4, $K$=8), HIS: ($B$=8, $K$=8), PABP: ($B$=4, $K$=4) and CAP: ($B$=2, $K$=7).

For each strategy proposed in Meier et al. (2021), as an ablation, we modify them to consider the likelihood of every token in the sequence when computing the score, rather than just the likelihood at the mutations (we denote these modified strategies with $'$). As a result, the summation in Equation (5) is modified from $\sum_{i:x^i_{\text{wt}} \neq x^i}$ to $\sum_i$.

For the first time, we demonstrate that whilst not all additional compute improves over the highly efficient wt-marginal strategy, the 'modulo' masking strategy (Johnson et al., 2024) outperforms all others with $n = 128$ or 512, but requires $K \cdot B$ times more forward passes, where $K = 4$ or 8 depending on dataset (specified in Section C.4).

Table 7: Masked PLM scoring strategies evaluated on five multi-mutant ProteinGym datasets, where $M$ is the number of mutations in a sequence, $B$ is the batch size, and $K$ is the masked-modulo constant (set to 4 or 8 depending on dataset).

| Scoring Function | Steps | Loss | ESM-1v | | | ESM-2 | | |
| --- | --- | --- | --- | --- | --- | --- | --- | --- |
| | | | $n = 32$ | $n = 128$ | $n = 512$ | $n = 32$ | $n = 128$ | $n = 512$ |
| wt-marginals | 1 | mse | 0.446 | 0.414 | 0.544 | 0.461 | 0.407 | 0.522 |
| Meier et al. (2021) | | ranking | 0.577 | 0.642 | 0.753 | **0.593** | **0.651** | 0.758 |
| masked-mt-marginals | $B$ | mse | 0.389 | 0.362 | 0.562 | 0.392 | 0.328 | 0.561 |
| Meier et al. (2021) | | ranking | 0.522 | 0.650 | 0.755 | 0.559 | 0.651 | 0.766 |
| masked-mt-marginals′ | $B$ | mse | 0.572 | 0.604 | 0.602 | 0.571 | 0.629 | 0.632 |
| | | ranking | 0.555 | 0.586 | 0.647 | 0.558 | 0.606 | 0.635 |
| mt-marginals | $B$ | mse | 0.214 | 0.291 | 0.533 | 0.264 | 0.330 | 0.530 |
| Meier et al. (2021) | | ranking | 0.351 | 0.578 | 0.754 | 0.398 | 0.591 | 0.750 |
| mt-marginals′ | $B$ | mse | **0.579** | 0.617 | 0.622 | 0.565 | 0.612 | 0.627 |
| | | ranking | 0.552 | 0.644 | 0.767 | 0.525 | 0.646 | **0.771** |
| masked-modulo | $K \cdot B$ | mse | **0.579** | 0.621 | 0.645 | 0.568 | 0.586 | 0.620 |
| Johnson et al. (2024) | | ranking | 0.529 | **0.654** | **0.769** | 0.534 | **0.661** | **0.774** |

## C.5 Compute requirements

All experiments were run on either A100 or H100 NVIDIA GPUs. Compute required for a single fine-tuning run varies based on the model, the length of the protein sequences, and the size of the dataset. We provide representative timings averaged over the 8 single mutant datasets for $n = 512$ in Table 8. Design experiments involved 10 rounds of fine-tuning and therefore required roughly ten times the computation of a single fine-tuning run.

During this research, the compute resources used were more than required for the experiments detailed in the paper, as additional compute was utilised for preliminary experimentation.

Table 8: Representative run times for fine-tuning ($n = 512$) averaged over 8 single-mutant datasets and across 3 seeds, on an H100 GPU.

| Model name | Time |
|---|---|
| ProteinNPT (MSAT) | 24 m |
| ProteinNPT (ESM-1v) | 34 m |
| ESM-1v (linear head, mse) | 35 m |
| ESM-1v (wt-marginals, rank) | 7 m |
| ESM-2 (linear head, mse) | 15 m |
| ESM-2 (wt-marginals, rank) | 4 m |
| PoET (linear head, mse) | 7 m |
| PoET (likelihood, rank) | 7 m |

## C.6 Zero-shot PLM performance

As discussed in **??** Result 2, the ranking-based fine-tuning performance of PoET is not attributed directly to higher zero-shot performance of the base PLM. We evaluate the zero-shot performance of the base models here, using the $n = 128$ test splits and report the Spearman correlation between likelihood scoring function and the fitness measurement. The MSA Transformer and ESM-1v 5 model ensemble zero-shot predictions are taken from Notin et al. (2023a) for our test splits. These zero shot predictions are provided as inputs to ProteinNPT (MSAT and ESM-1v versions) and the corresponding augmented embedding baselines.

Table 9: Zero-shot Spearman correlation on the $n = 128$ test splits for the base PLM models.

| Base Model | Singles | Multiples |
|---|---|---|
| MSA Transformer | 0.399 | 0.534 |
| ESM-1v (wt-marginals) | 0.384 | 0.425 |
| ESM-1v (5 model ensemble) | **0.437** | 0.392 |
| ESM-2 (wt-marginals) | 0.372 | 0.493 |
| PoET (likelihood) | 0.417 | **0.592** |

## C.7 Low-n Fitness Prediction with Ensemble Models

Table 10: Low-n fitness prediction Spearman results comparing the masked- and family-based MSA-ensemble models to their single model counterparts. Averaged over three seeds and 8 single mutant datasets (left) and five multi-mutant datasets (right).

| Model Name | Scoring Fn. | Loss | Single-mutants | | Multi-mutant | |
|---|---|---|---|---|---|---|
| | | | $n = 32$ | $n = 128$ | $n = 32$ | $n = 128$ |
| PNPT (MSAT) | - | mse | 0.420 | 0.532 | 0.511 | 0.696 |
| PNPT (MSAT) w/ dropout | - | mse | 0.421 | 0.532 | 0.512 | 0.696 |
| ESM-2 | wt-marginal | ranking | 0.455 | 0.530 | 0.593 | 0.651 |
| | | mse | 0.330 | 0.267 | 0.461 | 0.407 |
| | linear head | ranking | 0.307 | 0.411 | 0.447 | 0.648 |
| | | mse | 0.280 | 0.398 | 0.427 | 0.596 |
| ESM-2 ensemble | wt-marginal | **ranking** | **0.477** | **0.553** | **0.621** | **0.683** |
| | | mse | 0.347 | 0.335 | 0.507 | 0.440 |
| | linear head | ranking | 0.357 | 0.435 | 0.511 | 0.694 |
| | | mse | 0.342 | 0.428 | 0.505 | 0.658 |
| PoET | likelihood | ranking | 0.514 | 0.594 | 0.667 | 0.736 |
| | | mse | 0.409 | 0.378 | 0.601 | 0.583 |
| | linear head | ranking | 0.468 | 0.575 | 0.574 | 0.723 |
| | | mse | 0.445 | 0.554 | 0.582 | 0.715 |
| PoET MSA-ensemble | likelihood | **ranking** | **0.524** | **0.607** | **0.696** | **0.757** |
| | | mse | 0.412 | 0.390 | 0.623 | 0.609 |
| | linear head | ranking | 0.504 | 0.598 | 0.618 | 0.744 |
| | | mse | 0.486 | 0.591 | 0.632 | 0.736 |

### C.8 Additional sequence design recall curves

#### C.8.1 ESM-2 masked-ensembles

ESM-2 masked-ensembles comparing the wt-marginal scoring strategy fine-tuning via ranking loss to the linear regression head fine-tuned with MSE loss. AUC = area under the curve (higher is better). Each ensemble contains 5 members, with more details specified in Section A.3. Evaluated on 8 single mutant datasets (left) and 5 multi-mutant datasets (right).

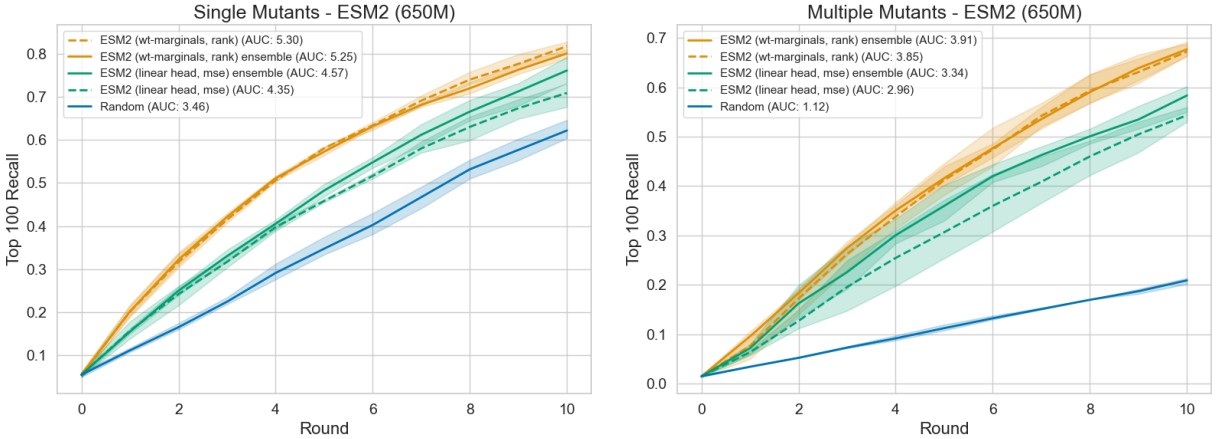

Figure 4: ESM-2 (650) masked-ensembles (left): single mutation datasets. (right) multiple mutation datasets.

#### C.8.2 Family-based PLMs

PoET MSA-ensemble comparing the likelihood fine-tuning via ranking loss to the linear regression head fine-tuned with MSE loss. AUC = area under the curve (higher is better). Each ensemble contains 5 members, with more details specified in Section A.2. Evaluated on 8 single mutant datasets (left) and 5 multiple mutant datasets (right).

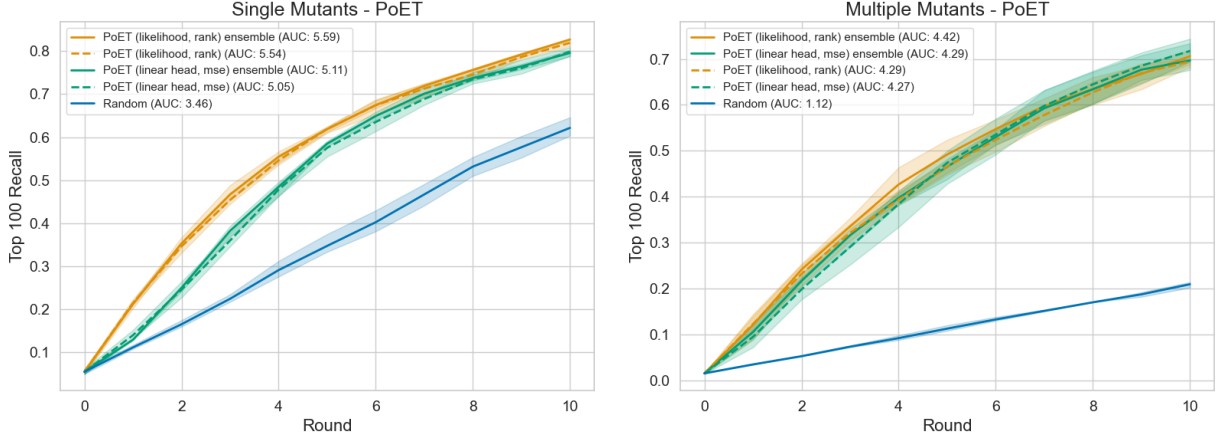

Figure 5: PoET MSA-ensembles (left): single mutants datasets. (right) multiple mutant datasets.

### C.8.3 ProteinNPT Baselines

ProteinNPT baseline methods taken from (Notin et al., 2023b). AUC = area under the curve (higher is better). Uncertainty is calculated using MC dropout, with more details specified in Section A.4. Evaluated on 8 single mutant datasets (left) and 5 multi-mutant datasets (right).

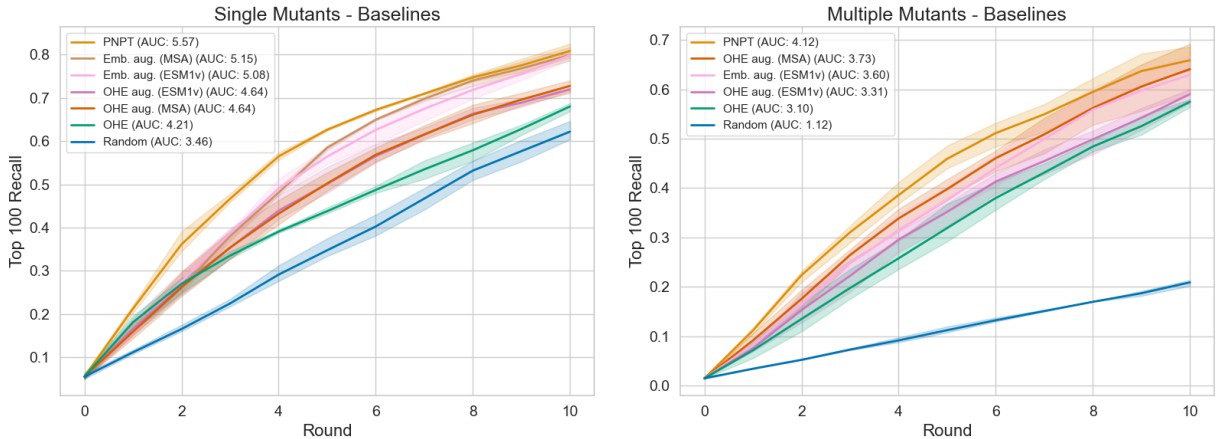

Figure 6: ProteinNPT baselines (left): single mutation datasets. (right) multi-mutation datasets.

### C.8.4 Single mutant dataset results

Each method is evaluated on each of the 8 single mutation datasets and each of the 5 multiple mutation dataset, repeated across 3 random seeds.

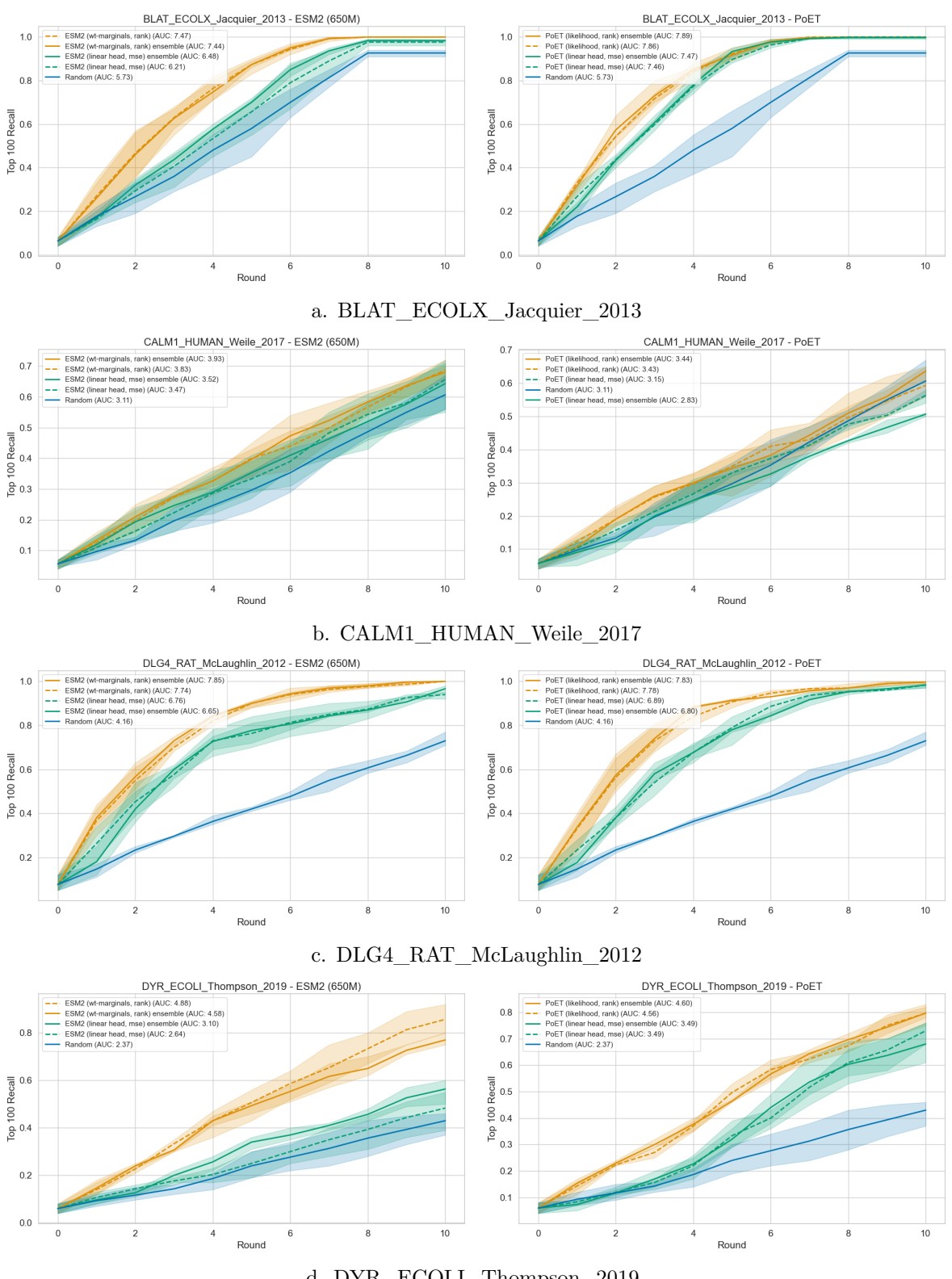

Figure 7: Sequence design top 100 sequences recall results on each of the single mutation datasets. Masked PLM ESM-2 and ESM-2 masked-ensembles (left), Family-based PoET and PoET MSA-ensemble (right).

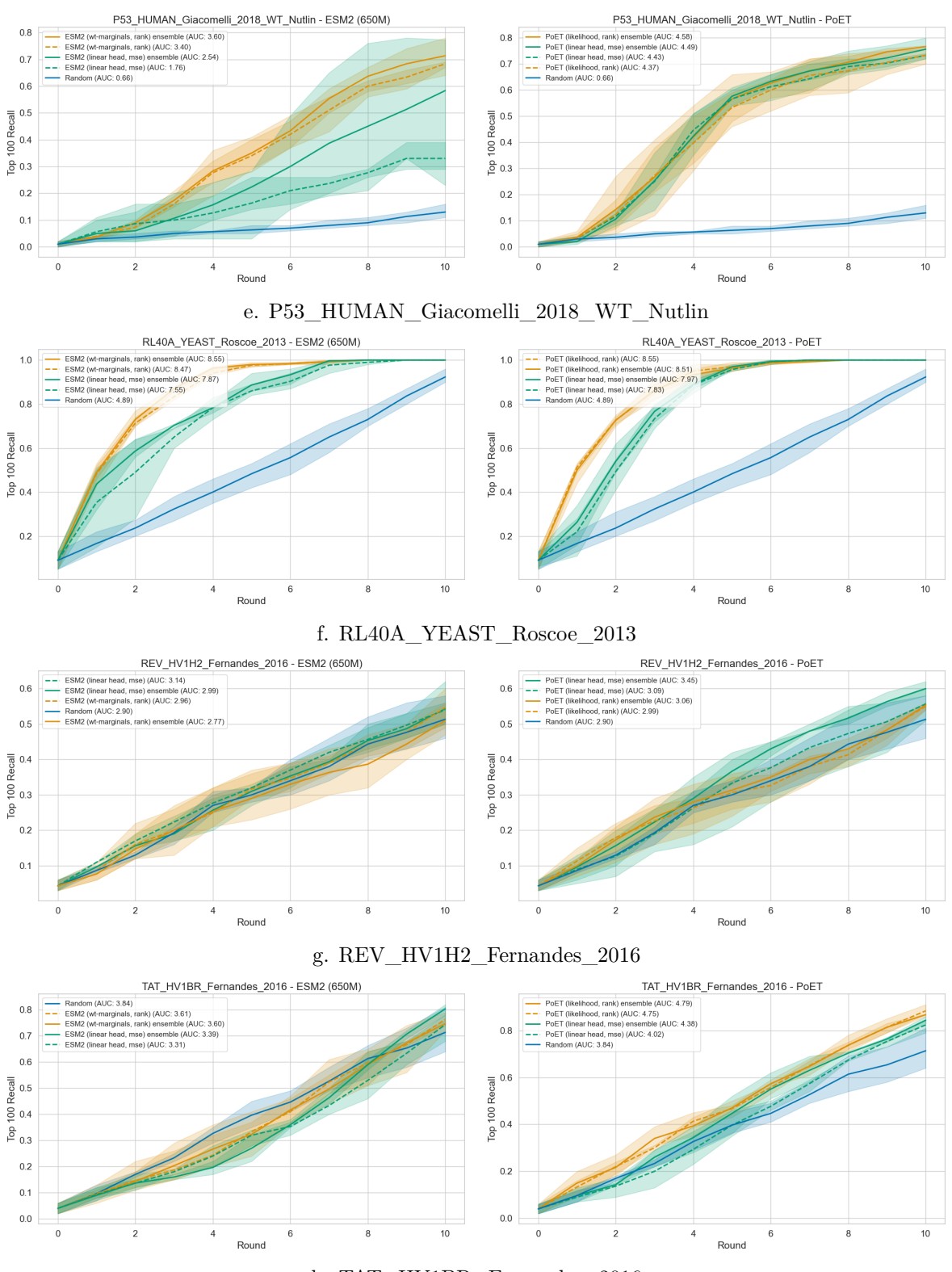

e. P53_HUMAN_Giacomelli_2018_WT_Nutlin

f. RL40A_YEAST_Roscoe_2013

g. REV_HV1H2_Fernandes_2016

h. TAT_HV1BR_Fernandes_2016

Figure 8: Sequence design top 100 sequences recall results on each of the single mutation datasets. Masked PLM ESM-2 and ESM-2 masked-ensembles (left), Family-based PoET and PoET MSA-ensemble (right).

## C.8.5    Multiple mutant dataset results

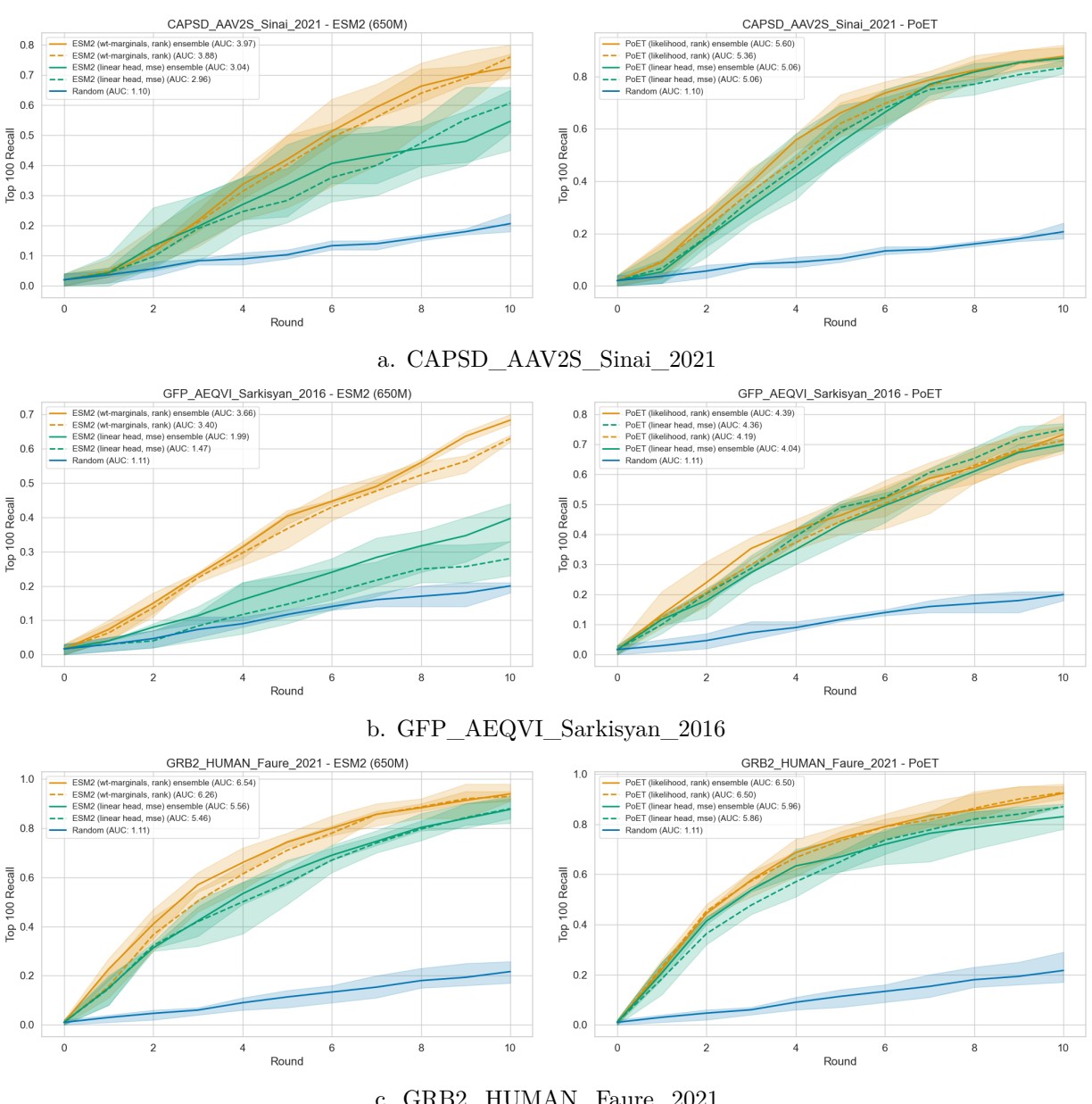

a. CAPSD_AAV2S_Sinai_2021

b. GFP_AEQVI_Sarkisyan_2016

c. GRB2_HUMAN_Faure_2021

Figure 9: Sequence design top 100 sequences recall results on each of the multiple mutation datasets. Masked PLM ESM-2 and ESM-2 masked-ensembles (left), Family-based PoET and PoET MSA-ensemble (right).

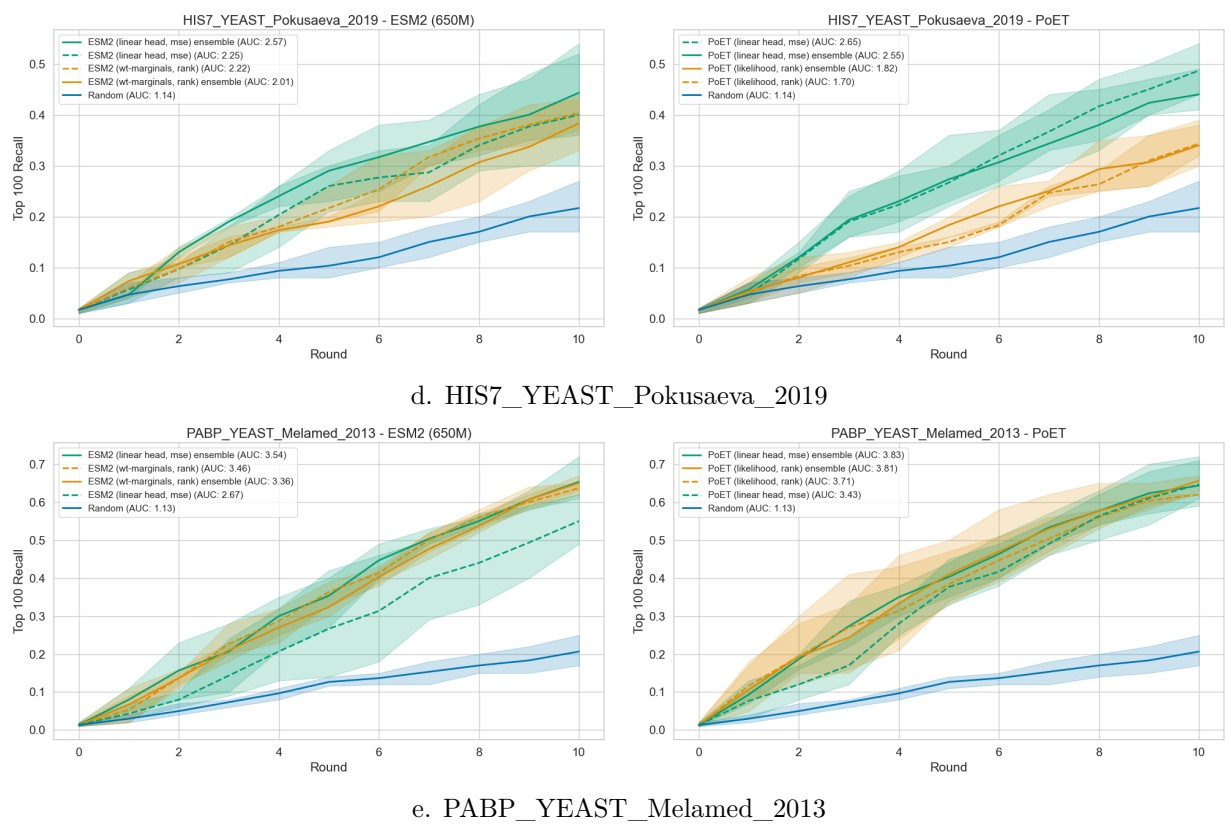

d. HIS7_YEAST_Pokusaeva_2019

e. PABP_YEAST_Melamed_2013

Figure 10: Sequence design top 100 sequences recall results on each of the single mutation datasets. Masked PLM ESM-2 and ESM-2 masked-ensembles (left), Family-based PoET and PoET MSA-ensemble (right).

## D    Broader Impact

We investigate methods for improved fine-tuning of PLMs to predict protein fitness. The development of these methods is motivated by the utility of the design of functional proteins for societally beneficial applications in pharma and biotechnology. Improved protein design strategies also lead to risks of enabling the design of harmful proteins. These risks apply more broadly to both the computational and experimental literatures on protein design rather than to our work in particular. However, we strongly urge the community to be informed about these risks and to give consideration to the broader impacts of any future uses and extensions of AI-based protein design tools.

Table 11: Sequence design full table of AUC results, across models, scoring functions, loss functions, ensembles, and baseline methods. AUC = area under the curve (higher is better) presented for both Top 30% recall (as per Notin et al. (2023b), and recall over the Top 100 sequences in the pool. Averaged over 8 single mutant datasets (left) and 5 multi-mutant datasets (right).

| | | | **Single Mutants** | |
|---|---|---|---|---|
| Model Name | Scoring Fn. | Loss | AUC (Top 100 Recall) | AUC (Top 30% Recall) |
| ESM2 (650M) | linear head | mse | 4.35 (0.07) | 4.40 (0.06) |
| | | ranking | 4.87 (0.18) | 4.61 (0.06) |
| | wt-marginals | mse | 4.52 (0.05) | 4.46 (0.03) |
| | | ranking | 5.30 (0.08) | 5.03 (0.03) |
| ESM2 (650M) ensemble | linear head | mse | 4.57 (0.11) | 4.60 (0.07) |
| | wt-marginals | ranking | 5.30 (0.06) | 5.12 (0.02) |
| PoET | linear head | mse | 5.04 (0.14) | 4.97 (0.02) |
| | | ranking | 5.53 (0.12) | 5.21 (0.03) |
| | likelihood | mse | 3.68 (0.14) | 3.70 (0.02) |
| | | ranking | 5.52 (0.08) | 5.21 (0.03) |
| PoET ensemble | linear head | mse | 5.11 (0.02) | 5.06 (0.01) |
| | | ranking | 5.60 (0.05) | 5.28 (0.04) |
| | likelihood | mse | 3.78 (0.09) | 3.74 (0.06) |
| | | ranking | 5.59 (0.05) | 5.26 (0.02) |
| PNPT (MSAT) + dropout | - | mse | 5.57 (0.02) | 5.10 (0.03) |
| Emb. aug. (ESM1v) | - | mse | 5.07 (0.11) | 4.95 (0.02) |
| OHE | - | mse | 4.19 (0.09) | 4.44 (0.04) |
| OHE aug. (MSA) | - | mse | 4.54 (0.04) | 4.68 (0.02) |
| OHE aug. (ESM1v) | - | mse | 4.52 (0.02) | 4.76 (0.03) |

| | | | **Multiple Mutants** | |
|---|---|---|---|---|
| Model Name | Scoring Fn. | Loss | AUC (Top 100 Recall) | AUC (Top 30% Recall) |
| ESM2 (650M) | linear head | mse | 2.96 (0.36) | 2.14 (0.03) |
| | | ranking | 3.58 (0.14) | 2.36 (0.01) |
| | wt-marginals | mse | 2.79 (0.29) | 2.01 (0.01) |
| | | ranking | 3.85 (0.25) | 2.58 (0.02) |
| ESM2 (650M) ensemble | linear head | mse | 3.34 (0.23) | 2.31 (0.03) |
| | wt-marginals | ranking | 3.91 (0.14) | 2.59 (0.03) |
| PoET | linear head | mse | 3.71 (0.32) | 2.24 (0.06) |
| | | ranking | 4.08 (0.51) | 2.44 (0.31) |
| | likelihood | mse | 1.58 (0.20) | 1.50 (0.03) |
| | | ranking | 3.73 (0.18) | 2.30 (0.02) |
| PoET ensemble | linear head | mse | 4.29 (0.20) | 2.78 (0.02) |
| | | ranking | 4.40 (0.22) | 2.78 (0.04) |
| | likelihood | mse | 1.73 (0.16) | 1.78 (0.08) |
| | | ranking | 4.45 (0.28) | 2.80 (0.03) |
| PNPT (MSAT) + dropout | - | mse | 4.12 (0.13) | 2.60 (0.03) |
| Emb. aug. (ESM1v) | - | mse | 3.61 (0.22) | 2.62 (0.03) |
| OHE | - | mse | 3.10 (0.22) | 2.33 (0.06) |
| OHE aug. (MSA) | - | mse | 3.73 (0.23) | 2.68 (0.03) |
| OHE aug. (ESM1v) | - | mse | 3.37 (0.28) | 2.47 (0.04) |

