# OpenReview forum: "Likelihood-based Fine-tuning of Protein Language Models for Few-shot Fitness Prediction and Design"
_TMLR — Rejected by TMLR_

### Review · Reviewer_pXGo · 2026-02-21

**Summary Of Contributions:**

This paper develops a simple, unified framework for likelihood-based fine-tuning for PLMs,

**Audience:**

Yes

**Audience Explanation:**

Yes — at least a meaningful subset of TMLR’s audience would likely be interested in this paper. Beyond its protein design application, the core contribution is methodological: it shows that likelihood-based ranking fine-tuning consistently outperforms regression-based adaptation of pretrained models in low-data regimes. This directly connects to broader ML themes such as preference learning, distributional fine-tuning, and efficient adaptation of foundation models — topics that are highly relevant to TMLR readers working on model alignment, representation learning, and ML for science.

**Broader Impact Concerns:**

not suitable.

**Claims And Evidence:**

No

**Claims Explanation:**

Strength:

- It extends ranking-based fine-tuning beyond unconditional AR PLMs to masked and family-based AR PLMs.

- The presentation is relatively good and clear. It tries to address a highly relevant pratical question for protein engineering. That is, how to adapt PLMs for a low-data regime.

Weaknesses and concerns.

- Pairwise ranking with BT has quadratic pair growth within a batch. The author fails to discuss the computational and memory implications. Also, it is necessary to discuss strategies like the hard-negative mining and pair subsampling, which may couple with the ranking loss.

- Uncertainty calibration is underexplored. Ensemble gains are modest or even negative. Authors hypothesize that this may be due to calibration issues, but do not provide quantitative calibration metrics.

- There are some mixed statements about which parameters are fine-tuned for PoET, which makes me confused. Namely, "all parameters" are fine-tuned vs. fine-tuning only a mapping layer with cached MSA encoding.

- Missing comparision with some related fine-tuning approaches for PLMs. Some scalable preference learning algorithms can also contextualize computational trade-offs of pairwise ranking.

- There are so many ranking losses, for instance, [1] Hinge/margin ranking loss, which is suitable for extramely low-N regimes and high label noise. [2] Soft margin, or smooth hinge, just what this paper used, but we can introduce a temperature-scale; [3] listwise cross entropy, which can lead to reliable totla ordering. [4]contrastive ranking losses such as InforNCE ranking, triplet loss. [5]some noise-robust ranking losses like Gaussian ranking, etc. This paper does not compare the advantage and disadvantage of those diverse ranking families, but only show it is better than the vanilla ESM-2

---------------
From my perspective, this paper lacks rigorous theoretical analysis and novel machine learning algorithms. Based on these factors, it does not meet the criteria for TMLR; it is more likely to be submitted to venues such as bioinformatics and other bio-oriented conferences.

**Requested Changes:**

Please address all my concerns in the previous weakness part. Most importantly, the authors need to convince me why this ranking loss is preferred to a wide range of other ranking loss choices. Otherwise, this is very much like a project technical report rather than a high-quality research paper. Also, there are some other PLMs not only containing ESM-2. It is good to see more experiments.

---

> ### Author Response · Authors · 2026-04-28
> **Response to review**
>
> Thank you for the very quick and thoughtful review!
>
> > Pairwise ranking with BT has quadratic pair growth within a batch. The author fails to discuss the computational and memory implications. Also, it is necessary to discuss strategies like the hard-negative mining and pair subsampling, which may couple with the ranking loss.
>
> We agree with the importance of clarifying this point but disagree with the reviewer’s assessment of the quadratic pair growth of pairwise ranking with BT. In fact, the main computational cost is (at worst) linear in the batch size B. This is because the model produces a scalar score for each item in the batch, and these B scores are then tiled to produce a B x B set of pairwise comparisons on which the loss is computed (see code implementation in Appendix A6). Thus the number of forward passes required grows linearly with B. Since the model forward pass on each of B sequences is much more expensive than the loss computation on BxB pairwise comparison scores, the effective cost is linear in B.
>
> We will comment on this after introducing the loss (Equation 2), by pointing to the implementation code in Appendix A6 and adding a comment on the linear way in which the number of forward passes grows with batch size.
>
> > Uncertainty calibration is underexplored. Ensemble gains are modest or even negative. Authors hypothesize that this may be due to calibration issues, but do not provide quantitative calibration metrics.
>
> We agree with this assessment and think that section 4.4 is currently written in a way that could be interpreted as positioning the paper as making a contribution towards uncertainty calibration. We do not want to make this claim, except in the weakest possible way, since we agree that our results do not justify conclusions about the success or otherwise of calibration. We propose to rewrite the first paragraph of this section (we'll post the updated paragraph in a separate comment below to respect character limits for this one).
>
> >There are some mixed statements about which parameters are fine-tuned for PoET, which makes me confused. Namely, "all parameters" are fine-tuned vs. fine-tuning only a mapping layer with cached MSA encoding.
>
> Thank you for identifying this, we agree with the assessment that this is confusing. The correct statement is in Appendix A2, but this main-text sentence is confusing in the case of PoET: “In the first case, we add a linear regression head to pooled embeddings extracted from the models, and fine-tune all parameters with an MSE loss (additional details in Appendix A.1.1).”
> We will switch to ‘fine-tune with an MSE loss’, and ‘(further details are provided in Appendices A.1.1 and A2)’
>
> > Missing comparision with some related fine-tuning approaches for PLMs. Some scalable preference learning algorithms can also contextualize computational trade-offs of pairwise ranking.
>
> > There are so many ranking losses, for instance, [1] Hinge/margin ranking loss, which is suitable for extramely low-N regimes and high label noise. [2] Soft margin, or smooth hinge, just what this paper used, but we can introduce a temperature-scale; [3] listwise cross entropy, which can lead to reliable totla ordering. [4]contrastive ranking losses such as InforNCE ranking, triplet loss. [5]some noise-robust ranking losses like Gaussian ranking, etc. This paper does not compare the advantage and disadvantage of those diverse ranking families, but only show it is better than the vanilla ESM-2.
>
> Such results would of course be interesting but we do not agree that they are required to support our core claims, which relate to the benefits of (a simple implementation of) ranking-based fine-tuning over the very widely used regression-based fine-tuning.
> We believe these results are of interest and value to the community. Investigation of different types of ranking losses is, we believe, tangential to the core contribution we are aiming to make with the paper. We hope that our paper could encourage further future work in this direction.
>
> We disagree with the claim that ‘we only show that it is better than the vanilla ESM-2’. We show that likelihood-based fine-tuning outperforms regression based fine-tuning for a given model, across model classes. We consider 3 models (ESM-1v, ESM-2 and PoET) representative of 2 model classes and see consistent gains. We also show improvements relevant to recent state-of-the-art methods on the ProteinGym datasets considered here - namely ProteinNPT and Kermut. We do not believe that the ranking loss experiments suggested would strengthen the extent to which we successfully support these claims.

---

> > ### Author Response · Authors · 2026-04-28
> > **Updated first paragraph of section 4.4**
> >
> > *Both masked and family-based autoregressive PLMs define distributions over sequence space conditioned on evolutionary context, such as a wild-type sequence or multiple sequence alignment. In the preceding sections, we effectively fixed the evolutionary context, by sampling a single input MSA with which to fine-tune family based models, and by fine-tuning masked language models using the full wild-type sequence as input. However, these models' ability to condition on evolutionary context naturally leads to the question of whether aggregating predictions across multiple related evolutionary contexts can yield improved predictions. We therefore extend our fine-tuning schemes to fine-tune ensembles of models from a single set of pretrained weights, but conditioning on distinct evolutionary contexts.*

---

> ### Author Response · Authors · 2026-05-01
> **Note on revision**
>
> We've uploaded a revision which addresses the concern around the lack of discussion of computational efficiency by adding the following to the background section where the Bradley-Terry formulation is introduced:
>
> > Importantly, despite this formulation’s reliance on pairwise comparisons, the computational cost associated with computing the loss on a batch of B inputs remains linear in B. This is because only B independent evaluations of the scoring function $s_\theta(x)$ are required to compute the B×B terms in the loss function in Equation 2. A reference implementation of the loss function illustrating this point is provided in Appendix A.6.
>
> The revision also incorporates the revised first paragraph for section 4.4 from above, and resolves the ambiguity around which PoET parameters that are fine-tuned, thanks for pointing this out.

---

### Review · Reviewer_qPQf · 2026-03-07

**Summary Of Contributions:**

This paper performs an empirical comparison between likelihood-based methods for protein fitness prediction (essentially fine-tuning a protein language model) vs training predictive models on embeddings from protein language model. They generally find that likelihood-based fine tuning performs better.

**Additional Comments:**

**Formatting**: a few changes would greatly improve the presentation of the paper:

1. In figure 1, put the datasets in the plot titles in addition to in the caption. The current presentation makes it hard to tell what changed between the top and bottom rows.
2. Please insert plots as vector graphics instead of raster images. This makes the text searchable. You can save as pdf with matplotlib.
3. Increase the tiny font sizes in some legends, particularly in the appendix. I recommend the tueplots package for formatting. Also, AI coding tools are really good at plot formatting. It should be a <30m job to make your plots look outstanding with AI if you put these critiques in with natural language.
4. You can't typeset scientific notation with math: $1e-5$ looks like $e-5\approx 2.7-5$. Write `1e-5` (`\texttt` in latex) or $10^{-5}$.

**Good job including a few things often omitted from papers!**

- Broader impact statement
- Measure of statistical variation in most tables / plots
- Compute requirements

**UCB**: the fixed $\lambda$ of 0.1 should not necessarily just be used for "consistency". $\lambda$ basically controls the explore-exploit tradeoff. The setting should depend on your time horizon. No need to change anything, but "consistency" seemed like a weak justification to me.

**Audience:**

Yes

**Audience Explanation:**

Protein design is a popular topic with clear practical importance to the pharmaceutical industry. I imagine that the results of this study will be informative for many in the field.

**Broader Impact Concerns:**

None. I appreciated the others adding a broader impact statement in the appendix.

**Claims And Evidence:**

Yes

**Claims Explanation:**

Yes: in section 5.2 they show likelihood-based fine-tuning outperforming regression on embeddings for many different models.

**Requested Changes:**

Aside from some minor formatting concerns (see "additional comments"), I had only 2 substantial concerns:

1. Code to reproduce this experimental work is not included. To make future research and analysis easier, please include all source code and raw results.
2. Although likelihood methods clearly outperform regression in the small data regime (e.g. n=24), in Fig 1-2 the differences between the methods grows smaller with larger n (e.g. n=512). It would be good for the authors to highlight this more explicitly- it is not emphasized at all in the summary of findings (at least as far as I could find). It also raises a question: would the performance trend reverse for larger n? Perhaps a study of n=1024 or 2048 would answer this. If that's not feasible, please directly address this as a limitation.

---

> ### Author Response · Authors · 2026-05-01
> **Response to review**
>
> Thanks for the review and the several helpful suggestions.
>
> In response to your main concerns:
>
> 1: A library will be open sourced to replicate the results of the paper shortly.
> 2: We agree it is important to draw attention to the role of n in the effectiveness of different fine-tuning schemes. In the second paragraph of Result 1 we attempt to do this already:
>
> "The performance gap narrows for all models as n increases, suggesting that larger datasets partially compensate for the suboptimal initialisation introduced by adding an untrained linear prediction head to pooled embeddings.”
>
> However, we do not currently address what happens at higher n. We feel this is justified by our explicit focus on the low-n setting and our identification of clear trends for various n <=512. Still, we agree that further experiments at higher n would of course be interesting, and will more explicitly acknowledge our focus on lower n as a limitation in the next version, while maintaining that it does not reduce the support for our core claims.
>
> In response to the requested changes, we've addressed the typesetting issue in the revision just uploaded, and will address the figure formatting issues in any final revision.

---

### Review · Reviewer_bzbm · 2026-04-16

**Summary Of Contributions:**

This paper studies how to best adapt existing protein language models when only a small number of experimental measurements are available, and shows that a particular kind of preference‑learning style fine‑tuning is a better default than standard regression approaches.

Strength:
Simple, unified likelihood-based ranking framework that works across masked and family-based PLMs with clear scoring rules.

Weakness:
The main text is very dense. long method and results sections with relatively few figures and almost all detailed quantitative results pushed to appendix tables. There is also no high-level method schematic or workflow diagram to visually summarise the unified ranking framework across PLM types, which makes the narrative feel heavier than it needs to be.

The results also feels limited to ProteinGym-style retrospective benchmarks and a small set of PLMs, so generality beyond these is uncertain. Also the ablations on why ranking helps (loss vs likelihood initialisation vs pair count) and on alternative scoring strategies, especially for multi-mutants feels limited.

The evidence is almost entirely Spearman for prediction + top‑100 AUC for design. There is no deeper analysis of score distributions, calibration, or alternative ranking/selection metrics.

**Audience:**

Yes

**Audience Explanation:**

It might be interesting to biology ML people and general representation learning audiences.

**Broader Impact Concerns:**

/

**Claims And Evidence:**

No

**Claims Explanation:**

Empirical shows consistent gains over regression and frozen-embedding SOTA in low‑n across multiple PLMs and ProteinGym landscapes. However, the paper relies almost entirely on Spearman correlation as the main metric, and higher correlation does not necessarily imply better practical performance for design (e.g., on the extreme high‑fitness tail). The evidence is not clear enough and needs a more detailed study.

**Requested Changes:**

1. the method section is very text‑heavy and need a high‑level schematic / block diagram. Given they unify masked, autoregressive, and MSA‑conditioned PLMs under one ranking framework, a single overview figure would materially improve clarity.
2. The Background feels somewhat trivial and can be shortened, or even move to the appendix to allocate space for figures and tables.
3. The whole paper can be condensed, especially on Introduction, Related work, and the background sections.
4. It reads weird as Results and then 5.2 Results sharing the same header name.
5. I recommend adding analysis of the score distributions (e.g., violin or histogram plots by fitness bin) to better understand how the proposed method separates high‑ vs low‑fitness sequences and whether its scores are meaningfully more useful for selection.

---

> ### Author Response · Authors · 2026-04-28
> **Response to review**
>
> Thanks for the thoughtful review which points out many interesting directions for extension of the work as well as improvement of the presentation.
>
> We respond to individual comments on weaknesses and strength of support for claims here, we will address the suggestions regarding presentation made in the 'requested changes' separately.
>
> > The results also feels limited to ProteinGym-style retrospective benchmarks and a small set of PLMs, so generality beyond these is uncertain.
>
> We want to defend the scope of the present study. ProteinGym is a standard benchmark setting, and we follow - though with variations that are discussed in the text - previous work published in prominent ML conference venues in our evaluation settings (such as ProteinNPT).
>
> > Also the ablations on why ranking helps (loss vs likelihood initialisation vs pair count) and on alternative scoring strategies, especially for multi-mutants feels limited.
>
> We accept that the results on alternative scoring strategies for multi-mutants are limited, but these are provided as an explorative analysis of certain patterns in the results: we are not claiming to have the optimal strategy for finetuning masked language models on multi-mutant data - indeed we explicitly note that multi-mutant data may be a setting where our choice of scoring function for masked language models underperforms. This is not an intrinsic failure of likelihood-based fine-tuning as evidenced by the strong performance of autoregressive models in these settings. Therefore we do not believe that extending these experiments would lend any additional support to our core claims. More broadly, we believe that the consistency of our results across representative models from two model classes and two in silico settings lend strong support to the claim that likelihood-based finetuning outperforms regression-based finetuning in typical settings. We considered it most important to demonstrate these results with multiple configurations of the fine-tuning setup applied to the same model in each case, because it is ultimately the particularity of the fine-tuning setup rather than the models themselves that our work addresses.
>
> > the paper relies almost entirely on Spearman correlation as the main metric, and higher correlation does not necessarily imply better practical performance for design (e.g., on the extreme high‑fitness tail). The evidence is not clear enough and needs a more detailed study.
>
> We want to push back in particular on this claim. We explicitly provide both supervised prediction results and in silico design results, thereby demonstrating that the conclusions made (using analysis of Spearman correlations) in the supervised prediction setting translate to the design setting, where the reviewer is correct to point out that only the high-fitness tail counts. While we agree that supervised spearman on its own leaves open the question of how the resulting differences in performance actually translate to a more practical setting, we have performed extensive design experiments to address precisely this question. We believe that the consistency of our results across supervised and design settings is strong evidence that the methods are effective.

---

> > ### Author Response · Authors · 2026-05-01
> > **Note on revision**
> >
> > We've uploaded a revised manuscript addressing the textual issues identified in the review. The previous background and related work contents have been contents. We did also add a couple of sentences on computational efficiency to the background in response to reviewer pXGo's comments. We think it is important to retain the background section in the main text to clarify the what things are taken directly from previous work.
> >
> > We have broken what used to be section 5 ‘Results’ into two separate sections: 5 ‘Low-n fitness prediction’ and 6. ‘Multi-round optimisation’ - thanks for pointing out the confusing structure here.
> >
> > We acknowledge the suggestions regarding figures/schematics and are deferring them for now to any final revision, we hope that this is reasonable given our comments above about the relevance of the design results in addressing the concern in requested change 5.

---

### Decision · Action_Editor_Qb5Z · 2026-05-27

**Recommendation:** Reject

**Additional Comments:**

I would expect the concerns of reviewer bzbm to be fully addressed before a resubmission.

**Audience:**

Yes

**Audience Explanation:**

All reviewers responded yes.

**Claims And Evidence:**

No

**Claims Explanation:**

There was disagreement among reviewers on this point, but 2/3 reviewers recommended rejection. One of the reviewers recommending rejection cited lack of novelty, which I am not taking into account in my recommendation. Instead, I am concerned that the concerns of Reviewer bzbm were not fully address in rebuttal. Specifically, a more detailed analysis of score distributions is still missing. Therefore I am recommending rejection.

**Resubmission Of Major Revision:**

The authors may consider submitting a major revision at a later time.